# Opto-RhoGEFs, an optimized optogenetic toolbox to reversibly control Rho GTPase activity on a global to subcellular scale, enabling precise control over vascular endothelial barrier strength

Eike K Mahlandt[1], Sebastián Palacios Martínez[1], Janine JG Arts[1,2], Simon Tol[2], Jaap D van Buul[1,2], Joachim Goedhart[1]*

[1]Swammerdam Institute for Life Sciences, Section of Molecular Cytology, van Leeuwenhoek Centre for Advanced Microscopy, University of Amsterdam, Amsterdam, Netherlands; [2]Molecular Cell Biology Lab at Dept. Molecular Hematology, Sanquin Research and Landsteiner Laboratory, Amsterdam, Netherlands

*For correspondence:
j.goedhart@uva.nl

Competing interest: The authors declare that no competing interests exist.

**Abstract** The inner layer of blood vessels consists of endothelial cells, which form the physical barrier between blood and tissue. This vascular barrier is tightly regulated and is defined by cell-cell contacts through adherens and tight junctions. To investigate the signaling that regulates vascular barrier strength, we focused on Rho GTPases, regulators of the actin cytoskeleton and known to control junction integrity. To manipulate Rho GTPase signaling in a temporal and spatial manner we applied optogenetics. Guanine-nucleotide exchange factor (GEF) domains from ITSN1, TIAM1, and p63RhoGEF, activating Cdc42, Rac, and Rho, respectively, were integrated into the optogenetic recruitment tool improved light-induced dimer (iLID). This tool allows for Rho GTPase activation at the subcellular level in a reversible and non-invasive manner by recruiting a GEF to a specific area at the plasma membrane, The membrane tag of iLID was optimized and a HaloTag was applied to gain more flexibility for multiplex imaging. The resulting optogenetically recruitable RhoGEFs (Opto-RhoGEFs) were tested in an endothelial cell monolayer and demonstrated precise temporal control of vascular barrier strength by a cell-cell overlap-dependent, VE-cadherin-independent, mechanism. Furthermore, Opto-RhoGEFs enabled precise optogenetic control in endothelial cells over morphological features such as cell size, cell roundness, local extension, and cell contraction. In conclusion, we have optimized and applied the optogenetic iLID GEF recruitment tool, that is Opto-RhoGEFs, to study the role of Rho GTPases in the vascular barrier of the endothelium and found that membrane protrusions at the junction region can rapidly increase barrier integrity independent of VE-cadherin.

## eLife assessment

This paper presents a **valuable** advance in the ability to manipulate the integrity of the barrier between endothelial cells. A wide range of data are presented, offering **convincing** support for the effectiveness of the method. This work is likely to attract a diverse audience of both cell biologists and researchers developing tools to manipulate cell and tissue function.

## Introduction

The endothelium lines the inner layer of blood vessels and functions as a physical barrier between tissue and blood. However, this barrier is permeable for gases, nutrients, ions, water, and also leukocytes, with which every organ in the human body is maintained. Despite these transport processes and even transmigration of leukocytes, the endothelium keeps the vascular barrier intact. Therefore, barrier strength, controlled by cell-to-cell contacts, needs to be tightly regulated. The dysregulation of these processes is linked to pathologies such as edema, arthritis, chronic inflammation, chronic bowel disease, cancer, infections, and other conditions (*Claesson-Welsh et al., 2021*).

Key elements defining the vascular barrier and permeability are the cell-cell junctions, the contact points between endothelial cells. Two types of junctions are especially important in the endothelium, namely adherens junctions, which are stable anchor points where cells attach to each other, and tight junctions, which seal off the two cell edges tightly. Regarding permeability, the transport of ions and small molecules below 800 Da happens via the tight junctions; transport of larger molecules and leukocytes across the endothelium is regulated by adherens junctions (*Duong and Vestweber, 2020*). Typically, linear adherens junctions show low leakage and are referred to as mature or stable adherens junctions (*Angulo-Urarte et al., 2020*). Homotypic VE-cadherin intercellular bonds are believed to mainly contribute to the vascular barrier strength (*Breviario et al., 1995*; *Gotsch et al., 1997*; *van Buul et al., 2002*). VE-cadherins are the typical vascular cadherins that bind homotypically to the VE-cadherin of the neighboring cell. They exist in a protein complex, including p120, alpha, beta and gamma catenin and this complex eventually connects to the actin network via α-catenin (*Angulo-Urarte et al., 2020*; *Dejana, 2004*). The actin cytoskeleton does not only define the cell morphology but also influences the organization of junctions (*Yamamoto et al., 2021*). Increased permeability and thereby reduced vascular barrier strength is associated with altered junction organization and gap formation (*Claesson-Welsh et al., 2021*).

On a molecular level, permeability is regulated by phosphorylation of junction proteins (*Claesson-Welsh et al., 2021*; *Vestweber, 2021*), which induce reorganization of the junctions. This phosphorylation is initiated by signaling molecules such as histamine, bradykinin, S1P, which are ligands for the GPCRs expressed in the endothelium. The GPCR-induced signaling cascades often involve Rho GTPases, which among others regulate the actin cytoskeleton and thereby junction organization as well. Other signaling modules involved are cAMP, NO, and SFK (*Claesson-Welsh et al., 2021*).

Here, we will focus on Rho GTPases and how they can influence endothelial cell-cell junction integrity. Rho GTPases are molecular switches that are 'ON' when they are bound to GTP and 'OFF' when they are bound to GDP. In the 'ON' state they initiate their signaling cascade, which among other options leads to the remodeling of the actin cytoskeleton. Rho GTPases are in turn regulated by Rho guanine-nucleotide exchange factors (GEFs), which facilitate the exchange of GDP to GTP, thereby they turn the Rho GTPase on. In contrast, G-protein activating proteins enhance the intrinsic Rho GTPase hydrolysis function to hydrolyze GTP to GDP and they thereby switch off.

Active Rho GTPases are typically located at the plasma membrane, inactive once can be bound by GDP dissociation inhibitors and localize in the cytosol. Out of the 20 human Rho GTPases, RhoA, Rac1, and Cdc42 are studied most abundantly. Rho activity is associated with cell contraction and actin stress fiber formation localized at rear edge of a migrating cell and decreased vascular barrier strength, for example, induced by histamine and thrombin (*Beckers et al., 2010*; *Hall, 1998*). Rac1 activity is associated with lamellipodia formation and cell spreading, and Cdc42 activity with formation of filopodia both Rac1 and Cdc42 localizing at leading edge of a migrating cell and promote increased vascular barrier strength, for example, by the stimulation with S1P (*Hall, 1998*; *Reinhard et al., 2017*). The polarization of the migrating cell shows that the activity of Rho GTPases is well defined in time and space (*Welch et al., 2011*). To further illustrate there is local Rho activity present at the cleavage furrow of a dividing cell coordinating a ring of contracting actin, limited in time to the duration of cell division (*Mahlandt et al., 2021*), local Rho activity was observed at the migration pore in the endothelium during leukocyte transmigration, probably restricting the pore size with an actin ring, limited in time to the duration of transmigration (*Heemskerk et al., 2016*), local Rac activity was observed in dorsal cell membrane ruffle (*Arts et al., 2021*). Additionally, Rho and Cdc42 waves were observed in *Xenopus* oocytes showing incredible local patterns that move quickly over time (*Moe et al., 2021*). However, Rho GTPase signaling can also occur in a global manner, for example during invagination in *Drosophila* embryo (*Izquierdo et al., 2018*).

It would be ideal to study the influence of Rho GTPase activity on the vascular barrier with a tool that activates Rho GTPases specifically, controllable on a global and subcellular level and in a precisely inducible and reversible manner. These requirements exclude some of the more traditionally performed experiments, such as overexpressing constitutively active or inactive mutants of different Rho proteins. With this method, cells can adapt to the increased level of Rho GTPase activity, it is inducible within hours and not reversible and no subcellular activation can be achieved. The famous experiment of microinjecting constitutively active Rho GTPases (*Hall, 1998*) shows the immediate cellular reaction induced by active Rho proteins. It is induced immediately but also lacks the subcellular activation and it is not possible to globally activate an entire monolayer of cells. Furthermore, there are many inhibitors and stimulators available to regulate Rho GTPase signaling, for example histamine, thrombin both Rho activators (*Mahlandt et al., 2021*), sphingosine-1-phosphate (S1P), a Rac, Rho and Cdc42 activator (*Reinhard et al., 2017*), bradykinin, a Cdc42 activator (*Martinez Quiles et al., 2001*). Most of these are ligands for GPCRs, meaning they may induce a branch of signaling cascades including the Rho GTPase but also other pathways. Some of these might be specific for one Rho GTPase but other may regulate multiple signaling pathways. They are usually added to the medium of cells and thereby act globally on all the cells, not allowing subcellular activation. They potentially need incubation time and are mostly irreversible unless an antagonist is available.

Another method is the rapamycin heterodimerization system. This system utilizes the GEF catalytic active DHPH domains, by recruiting them to the plasma membrane, the natural location of Rho GTPases, where GDP for GTP exchange takes place. This method requires two proteins that have low binding affinity in the absence of rapamycin and high binding affinity in its presence. One component called the FRB protein (**F**KB12 and **R**apamycin **B**inding domain) is targeted to the plasma membrane and functions as 'bait'. Its FRB-binding partner, FKBP12 (**FK**506 **B**inding **p**rotein with a mass of **12** kDa), is called the 'prey' and is fused to the active domain of a GEF that is present in the cytosol. Once rapamycin is added, it instantly binds to FKBP12. The FKPB12-rapamycin complex has a high binding affinity for FRB and by binding, it is relocated from the cytosol to the plasma membrane, that is, the location of FRB. Hence, the prey binds the bait, resulting in GEF activity at the plasma membrane and local activation of the matching Rho GTPase (*Rossman et al., 2005*). However, it needs to be stressed that rapamycin is not without side effects in a human cell, but in fact is involved in the mTOR pathway and hence may affect this pathway as well, upon exogenous addition.

To circumvent these issues, an optogenetic heterodimerization tool can be used with the same principle. This tool allows reversible light-induced activation on a scale from a global to subcellular level. Optogenetics is the genetic modification with light sensitive proteins. A number of optogenetic tools with the ability to recruit to the plasma membrane are available: BcLOV (*Glantz et al., 2018*), Cry2+CIB1 (*Kennedy et al., 2010*), improved light-induced dimer (iLID) (*Guntas et al., 2015*), enhanced Magnets (eMags) (*Benedetti et al., 2020*), PhyB/PiF (*Levskaya et al., 2009*), and more can be found on the OptoBase website (*Kolar et al., 2018*). For iLID and PhyB/PiF, it has been shown that the optogenetic recruitment of TIAM-1 and ITSN can be used to trigger Rho GTPase activity and thereby induce cell extension (*Guntas et al., 2015*; *Levskaya et al., 2009*). Also BcLOV has been used to manipulate Rho GTPase activity (*Berlew et al., 2022*) as well as CRY2/CIB1 (*Izquierdo et al., 2018*).

For this study, iLID and enhanced Magnets (eMags)-based optogenetic tools were chosen because they can be activated locally and globally, are fast inducible and reversible, relatively small in size, which is of advantage for lentiviral packing, do not require co-factor addition to the cell medium, and can be targeted to a cellular structure of choice. Cry2+CIBN was excluded because of its clustering properties and its relatively limited local activation ability (*Benedetti et al., 2018*). Phytochrome-based heterodimerization tools were excluded because of the necessity to add a co-factor. LOV domain-based tools require the co-factor flavin mononucleotide (FMN) or flavin adenine dinucleotide, which are present in cell growth medium. BcLOV was excluded because it intrinsically targets the plasma membrane, so in a follow-up the bait location could not be changed.

We aimed to optimize and test an optogenetic tool to activate Rho GTPases, reversibly, globally, and on a subcellular level, in endothelial cell monolayers to study vascular barrier strength. Therefore, kinetics and recruitment efficiency of iLID and eMags were compared. Subsequently, iLID was chosen to create optogenetically recruitable RhoGEFs (Opto-RhoGEFs) namely, TIAM1(DHPH), ITSN1(DHPH), and RhoGEFp63(DH), respectively, activating the Rho GTPases Rac, Cdc42, and Rho. The membrane

tag of the original iLID was changed to an optimized anchor. In addition, we modified the sequence of the domains to SspB, tag, GEF to simplify the exchange of GEF and genetically encoded tag. A set of plasmids with different fluorescent tags was created for more flexibility in co-imaging. Endothelial cell lines, stably expressing Opto-RhoGEFs, were generated and showed precisely activatable and reversible control over vascular barrier strength, measured in real time using ECIS (electrical cell-substrate impedance sensing) technology. The use of a VE-cadherin blocking antibody thereby functionally blocking VE-cadherin revealed that the Opto-RhoGEF TIAM1 and ITSN1 promoted endothelial barrier strength in a VE-cadherin-independent manner. Detailed imaging suggested that the increase in barrier is induced by an increase in cell-cell membrane overlap at junction regions. These data show the potential of Opto-RhoGEFs to change the cell morphology, that is contraction and extension, globally as well as locally.

## Results

### Characterizing and optimizing an optogenetic RhoGEF recruitment tool

To study the role of Rho GTPases in an endothelial cell model in a time and space defined manner, mimicking the natural Rho GTPase activity, heterodimerization tools for guanine exchange factor recruitment were compared. Targeting of GEFs to the plasma membrane is an effective way of activating Rho GTPases. The targeting can be achieved by light or chemical induction of heterodimerization. Here, we compared optogenetic heterodimerization tools to the chemical heterodimerization system that uses rapamycin. The optogenetic heterodimerization tools 'iLID' (*Guntas et al., 2015*) and 'eMags' (*Benedetti et al., 2020*) were selected because of their fast on/off kinetics, size, potential to be targeted to other subcellular locations in the cell rather than the plasma membrane, using FMN as co-factor, which is naturally present in human cells, and potential for local activation. HeLa cells were used for the tool optimization because of easier handling and higher transfection rate in comparison to endothelial cells.

First, the on/off kinetics and the recruitment efficiency of the recruitment systems were compared in HeLa cells, expressing either the rapamycin system, Lck-FRB T2098L-mTurquoise-IRES-sYFP2xNES-FKBP1, or the iLID system, mVenus-iLID-CaaX and SspB-mScarlet-I, or the eMags system, eMagA-EGFP-CaaX and eMagB-TagRFP, stimulated either with 488 nm laser light or the chemical rapamycin (*Figure 1A*). The $t_{1/2}$ ON and OFF kinetics were obtained by fitting a mono-exponential model to the data (*Figure 1—figure supplement 1A and B*). eMags showed the shortest $t_{1/2}$ ON kinetics with 3.97 s and the fastest $t_{1/2}$ OFF kinetics with 18.6 s. However, the mean efficiency of the recruitment measured by normalized membrane to cytosol intensity ratio was with 1.5 the lowest. The iLID system showed comparable $t_{1/2}$ ON kinetics of 5.14 s and $t_{1/2}$ OFF kinetics of 20.3 s and it had a higher mean recruitment efficiency. The rapamycin system showed a longer $t_{1/2}$ ON kinetics of 43.1 s, which may partly be explained by the different type of activation, as the rapamycin was added to the medium and needed to diffuse into the cell. The $t_{1/2}$ OFF kinetics was not determined as rapamycin-induced recruitment is irreversible at the time scales used. The rapamycin system showed the highest recruitment efficiency with a normalized membrane to cytosol intensity ratio above 3.

To test the ability to activate a signaling pathway through recruitment of a protein to the plasma membrane, inter-SH2 (iSH2) from the p85 domain of phosphatidylinositol 3-kinase (PI3K) was applied in combination with the Akt-PH PIP3 biosensor. iSH2 interacts with endogenous p110 catalytic subunit of PI3K, when recruited to the plasma membrane, and produces phosphatidylinositol 3,4,5-triphosphate $(PI(3,4,5)P_3)$, which can be measured with the biosensor. The sensor consists of the pleckstrin homology (PH) domain of Akt1 and mCherry (*Kontos et al., 1998*). It binds $PI(3,4,5)P_3$ and thereby localizes to the plasma membrane, which can be measured as an intensity change. The intensity change of Akt-PH at the plasma membrane upon iSH2 recruitment was compared for the iLID, eMags, and the rapamycin system (*Figure 1B and C*). The rapamycin system showed the highest sensor localization at the plasma membrane with a median of 14% change in the membrane to cytosol intensity ratio and iLID showed a median change of 6.4%. Unexpectedly, activation of eMags showed no change in the intensity of the $PI(3,4,5)P_3$ biosensor. This finding may be explained by the low recruitment efficiency observed for eMags (*Figure 1—figure supplement 1C*). This low recruitment efficiency in turn influences the biosensor response, since there appeared to be a correlation between high recruitment efficiency of the iSH domain and relocation of the biosensor (*Figure 1—figure supplement 1D*).

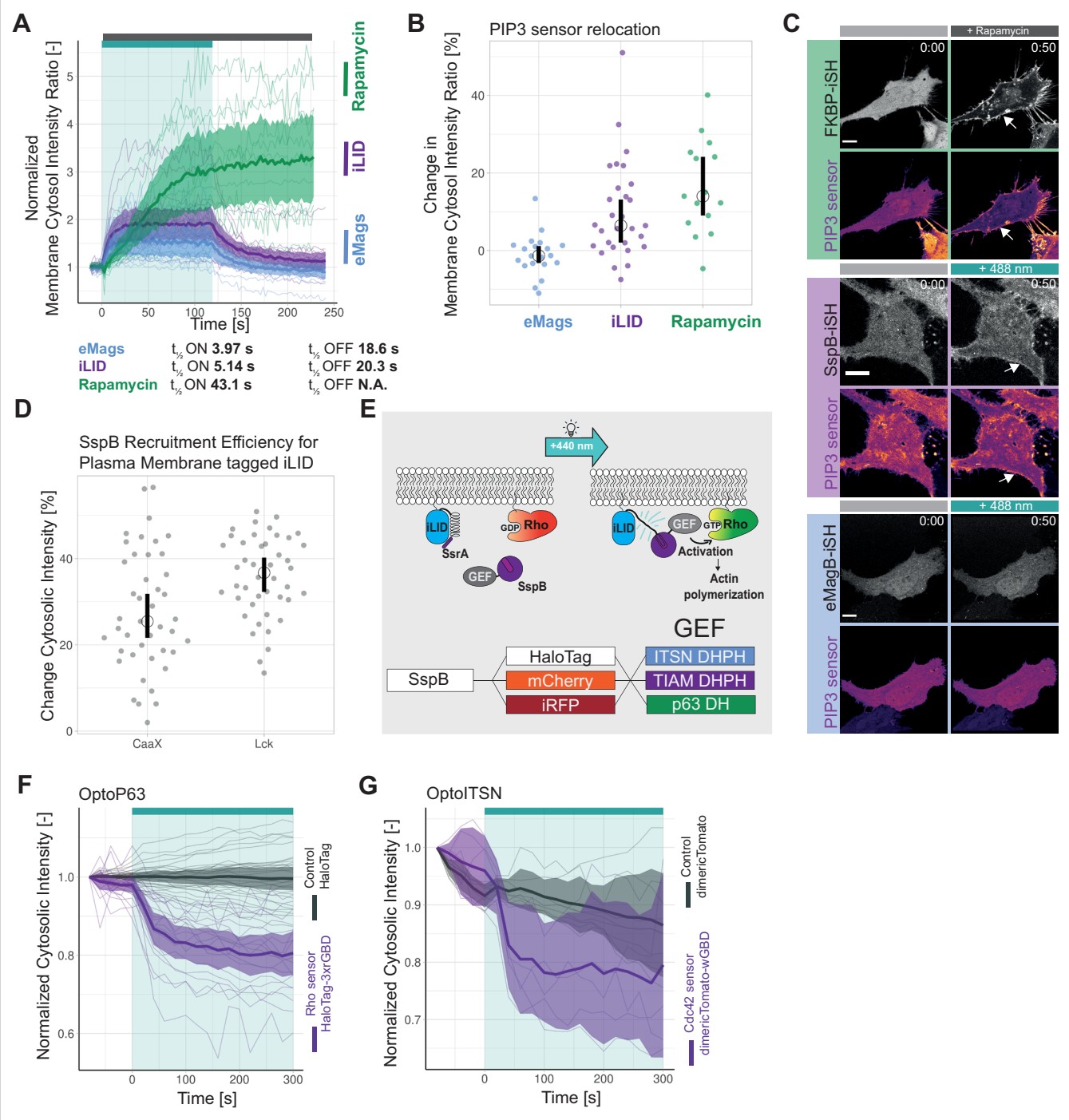

**Figure 1.** Optimization of an optogenetic recruitment tool. (**A**) Normalized membrane to cytosol intensity ratio for the rapamycin system (green) HeLa cells expressing Lck-FRB-T2098L-mTurquoise-IRES-sYFP2xNES-FKBP12, for the improved light-induced dimer (iLID) system (purple) expressing Venus-iLID-CaaX and SspB-mScarlet-I and for the enhanced Magnets (eMags) system (blue) expressing eMagA-EGFP-CaaX and eMagB-tagRFP stimulated with either 100 nM rapamycin (indicated by gray bar) or 488 nm laser light (indicated by cyan bar). The thin lines represent measurements of single cells, the thick lines represent the mean, and the ribbon represents the 95% confidence interval. The number of cells per condition is: iLID n=17, eMags n=12, rapamycin n=11. The data is from two biological replicates based on independent transfections. (**B**) Change in membrane to cytosol intensity ratio as percentage for the localization of the PIP3 sensor. All Hela cells were expressing the PIP3 location sensor mCherry-Akt-PH and for the iLID system (purple) Venus-iLID-CaaX, iSH-iRFP-SspB, for the eMags system (blue) eMagA-eGFP-CaaX and eMagB-iSH-iRFP670 and for the rapamycin system (green) Lck-FRB-mTurquoise2 and mNeonGreen-FKBP12-iSH. Stimulated with either 100 nM rapamycin or 488 nm laser light each frame. Comparing the ratio of membrane over cytosol intensity of the PIP3 sensor for 0 s pre-activation and 50 s of activation. Each dot represents an individual cell. The median of the data is shown as a black circle and the 95% confidence interval for each median, determined by bootstrapping, is indicated by the bar.

*Figure 1 continued on next page*

*Figure 1 continued*

The number of cells per condition is: iLID = 29, eMags = 20, rapamycin = 15. (**C**) Representative confocal images for the in B described conditions: HeLa cells expressing the eMag, iLID and rapamycin recruitment system and the PIP3 biosensorThe PIP3 sensor intensity is depicted with the mpl-inferno look up table, where brighter colors represent higher fluorescent intensities. Scale bars: 10 µm. (**D**) Change in cytosolic intensity in percentage for HeLa cells expressing SspB-mScarlet-I and either Lck-mTurquoise2-iLID or Venus-iLID-CaaX measured 20 s after stimulation with 1 pulse of 440 nm laser light at 20% for 1 s. Images acquired at a spinning disk microscope. Each dot represents an individual cell. The median of the data is shown as a black circle and the 95% confidence interval for each median, determined by bootstrapping, is indicated by the bar. The number of cells per condition is: CaaX = 43, Lck = 44. The data is from three biological replicates based on independent transfections. (**E**) Schematic of the light-induced heterodimerization of iLID. Upon photo-activation the Ja helix in the iLID unfolds, SsrA becomes available for binding by SspB which is recruited from the cytosol to the location of SsrA, which is localized at the plasma membrane. The guanine-nucleotide exchange factor (GEF) fused to the SspB activates the Rho GPTase at the plasma membrane, which is binding GTP and thereby activates its signaling cascade. That results in the remodeling of the actin cytoskeleton and change in cell morphology. The combination of fluorescent markers and GEFs fused to SspB in this study is indicated in the bottom. (**F**) Normalized cytosolic intensity for the HaloTag-3xrGBD Rho sensor (purple) or control HaloTag (gray) stained with JF635 nm upon the photo-activation (indicated by cyan bar) of SspB-mCherry-p63RhoGEF(DH), expressed in HeLa cells together with Lck-mTurquoise2-iLID. Thin lines represent individual cells, thick lines represent the mean, and ribbons represent their 95% confidence interval. The number of cells per condition: Rho sensor HaloTag-3xrGBD = 18, Control HaloTag = 30. The data is from two biological replicates based on independent transfections. (**G**) Normalized cytosolic intensity for the dimericTomato-wGBD Cdc42 sensor (purple) or control dimericTomato (gray) upon the photo-activation (indicated by cyan bar) of SspB-HaloTag-ITSN1(DHPH) stained with JF635 nm, expressed in HeLa cells together with Lck-mTurquoise2-iLID. Thin lines represent individual cells, thick lines represent the mean, and ribbons represent their 95% confidence interval. The number of cells per condition: Cdc42 sensor dimericTomato-wGBD=6, Control dimericTomato = 7. The data is from two biological replicates based on independent transfections.

The online version of this article includes the following figure supplement(s) for figure 1:

**Figure supplement 1.** Optogenetic tool setup.

At this point the iLID system was chosen, because the recruitment efficiency was robust and higher than for eMags and kinetics, even though slightly slower, were in the same range as for the eMags. Although the rapamycin system showed the highest recruitment efficiency and thereby the larger biological response of the biosensor, it lacks the reversibility and subcellular activation of an opto-genetics system. However, should these properties not be required in an experiment, then the rapa-mycin is a robust and efficient heterodimerization tool.

Next, the membrane tag of iLID was tested to improve recruitment efficiency (*Figure 1D*). As suggested in a previous study, an N-terminal membrane tag is more beneficial than a C-terminal tag (*Natwick and Collins, 2021*). The conventional C-terminal CaaX tag was compared to the N-terminal Lck tag. iLID tagged to the plasma membrane with Lck showed at higher median change in cytosolic intensity with 36.7%. The mean change in cytosolic intensity for the CaaX-tagged iLID was 25.5%; also its variance was larger compared to the Lck tag. Thus, iLID tagged with Lck showed improved recruitment efficiency, when compared with the CaaX box. Therefore, Lck-mTurquoise2-iLID was used as a bait in the remainder of the work.

To create a set of Opto-RhoGEFs with flexibility in the spectral window, we tagged SspB, serving as a 'prey', with the HaloTag, mCherry, or iRFP670. Three GEFs were selected, ITSN1, TIAM1, and RhoGEFp63, which are known to specifically activate respectively Cdc42, Rac, and Rho and their isoforms (*Figure 1E*). Their catalytic active DHPH domains were used for ITSN1 and TIAM1 (*Reinhard, 2019*, *Reinhard et al., 2021*). In case of p63 only the DH domain was used, because the PH domain of p63 inhibits the GEF activity (*van Unen et al., 2015*; ). Throughout the following text, the co-expression of Lck-mTurquoise2-iLID with either SspB-fluorescent tag-ITSN1(DHPH), SspB-fluorescent tag-TIAM1(DHPH), or SspB-fluorescent tag-RhoGEFp63(DH) will be referred to as OptoITSN, OptoTIAM, and OptoP63.

To test the specific activation of the Rho GTPases by Opto-RhoGEFs, single-color relocation sensors were applied, as they are compatible with the blue/green light absorption spectrum occupied by the LOV domain. The Rho sensor HaloTag-3xrGBD showed a clear relocation to the plasma membrane upon photo-activation of OptoP63, measured by fluorescent intensity decrease in the cytosol (*Figure 1F*). The recruitment of OptoP63 is shown in *Figure 1—figure supplement 1E*. The Cdc42 sensor dimericTomato-wGBD showed relocation to the plasma membrane upon photo-activation of OptoITSN (*Figure 1G*). The recruitment of OptoITSN to the plasma membrane and consequently reduced presence in the cytosol is shown in *Figure 1—figure supplement 1F*. No Rac1 sensor was available with a read-out that is compatible with optogenetics and with a dynamic range sensitive enough to measure the Rac1 activation upon OptoTIAM recruitment. TIAM1 has been

identified as a specific Rac GEF (*Müller et al., 2020*) and we observed for Rac activity expected cell spreading (Figure 4). The location-based sensors indicated that upon photo-activation OptoP63 triggered Rho activation at the plasma membrane, whereas OptoITSN triggered Cdc42 activity at the plasma membrane.

## Influence of Opto-RhoGEFs on permeability and barrier strength in endothelial cell monolayers

Endothelial cells line blood vessels and form the barrier between blood and tissue, but this barrier is semi-permeable, for example, for nutrients, oxygen, and leukocytes. The barrier function can be studied in a cell culture model, where endothelial cells grow in a monolayer. This model was used to test the effect of Opto-RhoGEFs on the vascular barrier, using several functional read-outs. To study the endothelial monolayer barrier function, endothelial cell lines, stably expressing Opto-RhoGEFs, were generated. Therefore, cord blood outgrowth endothelial cells (cBOECs) were treated with lentivirus to stably express Lck-mTurquiose2-iLID and either SspB-HaloTag-TIAM1(DHPH), SspB-HaloTag-ITSN1(DHPH), or SspB-HaloTag-p63RhoGEF(DH).

These experiments required photo-activation of large fields of endothelial monolayers, which could not be achieved at the microscope, as only a limited field of view can be illuminated. Therefore, a commercially available LED strip was used. The blue light setting on the LED strip had an emission peak ranging from 450 nm to 500 nm wavelength (*Figure 2—figure supplement 1A*). The light power density for the blue LED in its brightest setting was $9.4 \times 10^{-2}$ W/m$^2$, in comparison to the 442 nm laser at 1% intensity it was $8.58 \times 10^6$ W/m$^2$ and to the 488 nm laser, at 20% laser power, 1% intensity, it was $1.7 \times 10^7$ W/m$^2$. These values are not directly comparable to the photo-activation, as the LEDs illuminated the whole field of view continuously and the lasers scanned the area unidirectional at 400 Hz at the set frame interval. To test if the blue LED could be used for photo-activation, the same set of HeLa cells expressing Lck-mTurquoise2-iLID and SspB-mScarlet-I was exposed to either the blue LED light or laser light (*Figure 2—figure supplement 1B*). The photo-activation with blue LED light was sufficient to induce the heterodimerization of the iLID system. However, the photo-activation with laser light resulted in a more complete heterodimerization. Subsequently, LEDs were set to the brightest setting and used for photo-activation of full endothelial monolayers in incubators.

A permeability assay was performed, to study if the photo-activation of OptoTIAM changed the endothelial monolayer permeability. Therefore, BOECs stably expressing Lck-mTurquoise2-iLID and SspB-HaloTag-TIAM1(DHPH) or control cells expressing only Lck-mTurquoise2-iLID were grown into a monolayer in a Transwell dish on a polycarbonate membrane. This membrane, with the monolayer of endothelial cells on it, separated the top and the bottom compartment. To measure permeability of the endothelial cell monolayer, fluorescent dye, namely FITC, coupled to dextran of different kDa sizes was added to the top compartment and the fluorescence intensity is measured in the bottom compartment. Here, the permeability was measured after 10 min of incubation with FITC dextran either in the presence of blue light or in the dark. For three different particle sizes, namely 0.3 kDa, 10 kDa, 70 kDa, 10 min blue light-activated OptoTIAM endothelial cell monolayers showed the lowest permeability in comparison to OptoTIAM-expressing endothelial cell monolayers that were kept in the dark or photo-activated control monolayers (*Figure 2A*). The photo-activation of OptoTIAM cells seemed to decrease the permeability of the monolayer.

To study the influence of Opto-RhoGEFs on the endothelial monolayer barrier strength in real time, resistance was measured using ECIS technology. Therefore, BOECs stably expressing Lck-mTurquoise2-iLID and either SspB-HaloTag-TIAM(DHPH), SspB-HaloTag-ITSN(DHPH), or SspB-HaloTag-p63(DH) were grown as monolayers on ECIS electrode arrays. BOECs, only expressing Lck-mTurquoise2-iLID, were used as control. Cells were stimulated in the ECIS incubator with the blue LED light, as described above. OptoTIAM-expressing cells showed an increase in resistance after 1 min of activation with blue light, which is fully reversible within seconds. When activating for 10 min, the increase in resistance reached a maximum level. To study fatigue a sequential activation of 15 min was performed three times. Interestingly, during this period, the cellular system did not show any fatigue. Even when the endothelial cells were activated for 2 hr, the maximum resistance level did not show any drop, but as soon as photo-activation was stopped, the resistance decreased drastically within seconds. The control cells, only expressing the iLID part of the system, did not respond to the blue light with a change in resistance (*Figure 2B*, *Figure 2—figure supplement 3A*, *Figure 2—figure supplement 4A*). The

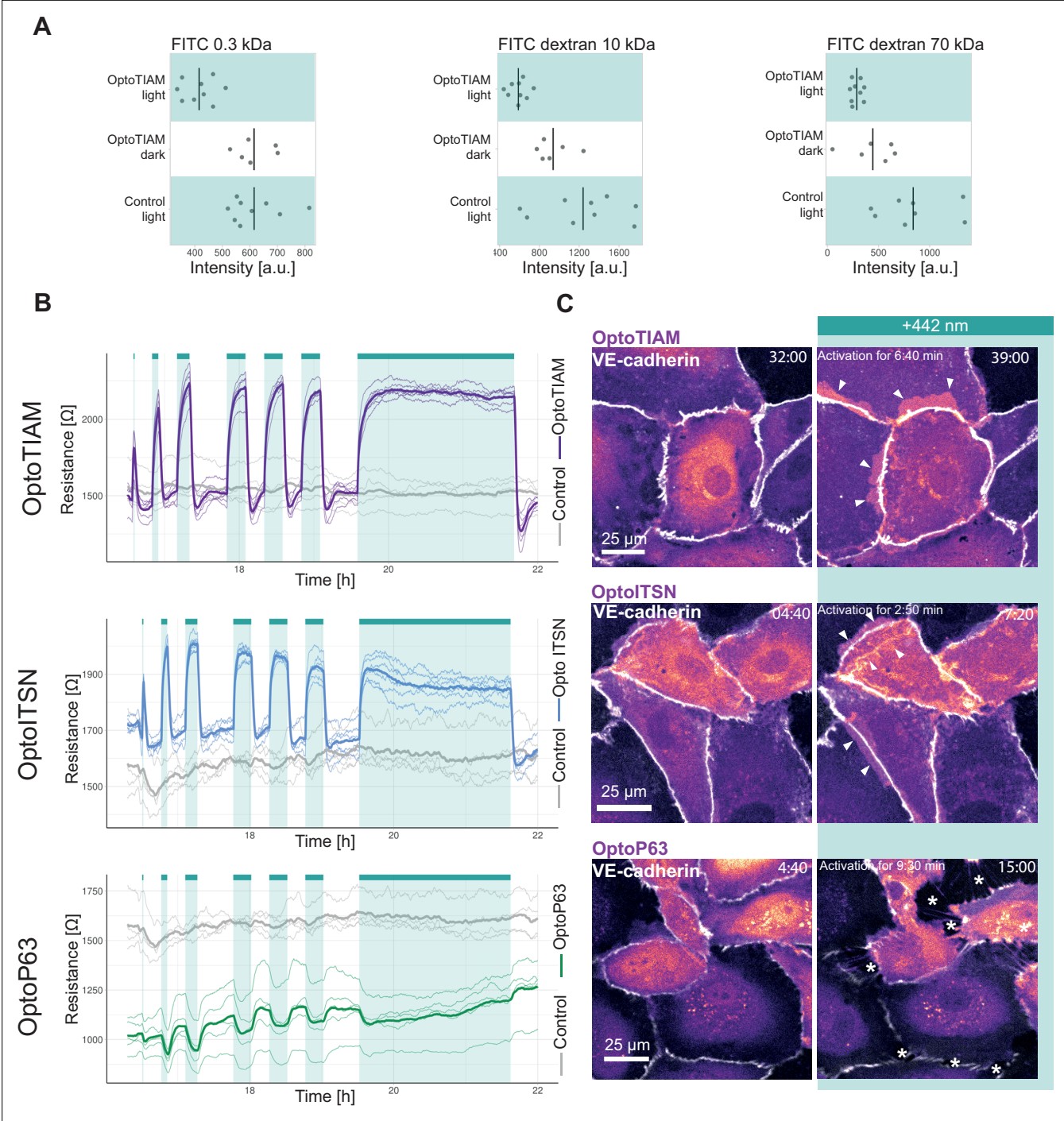

**Figure 2.** Photo-activation of optogenetically recruitable RhoGEFs (Opto-RhoGEFs) controls permeability and vascular barrier strength.
(**A**) Fluorescence intensity measured in a transwell assay for a monolayer of blood outgrowth endothelial cells (BOECs) stably expressing Lck-mTurquoise2-iLID, solely as a control, and SspB-HaloTag-TIAM1(DHPH) treated with FITC 0.3 kDa, FITC dextran 10 kDa, and FITC dextran 70 kDa. Photo-activated with blue LED light for 10 min as indicated by cyan background in the graph or kept in the dark indicated by white background. Dots represent individual transwell dishes. The black bar indicates the mean. The number of transwell dishes per condition is: Control Lck-iLID=9, OptoTIAM dark = 6, OptoTIAM light = 9. The data is from three experiments. (**B**) Resistance of a monolayer of BOECs stably expressing Lck-mTurquoise2-iLID, solely as a control (gray), and either SspB-HaloTag-TIAM1(DHPH)(purple)/ITSN1(DHPH) (blue) or p63RhoGEF(DH) (green) measured with electrical cell-substrate impedance sensing (ECIS) at 4000 Hz, representing paracellular permeability, every 10 s. Cyan bars indicated photo-activation with blue LED light (1 min, 5 min, 10 min, 3×15 min, 120 min). Thin lines represent the average value from one well of an 8W10E PET ECIS array. Thick lines represent the mean. The number of wells per condition was: OptoTIAM = 6, OptoITSN = 6, OptoP63=6, control = 4. (**C**) Representative zoom-ins from confocal

*Figure 2 continued on next page*

*Figure 2 continued*

microscopy images of a BOEC monolayer stably expressing Lck-mTurquoise2-iLID (not shown) and either SspB-HaloTag-TIAM1(DHPH)/ ITSN1(DHPH) or p63RhoGEF(DH) stained with JF552 nm dye (LUT = mpl-magma, bright colors indicating higher intensity). Additionally, VE-cadherin was stained with the live labeling antibody Alexa Fluor 647 Mouse Anti-Human CD144 (white). Scale bars: 25 µm. Times are min:s from the start of the recording. Cyan bar indicates 442 nm photo-activation. Arrows indicate overlap and protrusions. Asterisks indicate holes in monolayer. Whole field of view is shown in *Figure 2—figure supplement 2*.

The online version of this article includes the following figure supplement(s) for figure 2:

**Figure supplement 1.** Photo-activation with blue LED.

**Figure supplement 2.** Global photo-activation of endothelial cell monolayer.

**Figure supplement 3.** Replicate electrical cell-substrate impedance sensing (ECIS) assay and time-lapse microscopy.

**Figure supplement 4.** Overview of entire time course electrical cell-substrate impedance sensing (ECIS) assay.

resistance of the OptoITSN-expressing endothelial cells showed a similar pattern in comparison to the OptoTIAM cells but with a smaller amplitude, indicating that Cdc42 activation did not increase endothelial resistance to the same extent as Rac did. However, this monolayer reached maximum resistance at approximately 5 min. When OptoITSN cells were photo-activated for longer than 10 min the resistance started to decrease slightly but did not return to baseline. During the sequential three times 15 min photo-activation, the resistance increase was slightly smaller with each activation for the OptoITSN cells (*Figure 2B*, *Figure 2—figure supplement 3A*, *Figure 2—figure supplement 4A*).

OptoP63 cells showed a dip in resistance when activated for 1 min. The resistance of the monolayer returned to baseline within minutes. When activated for longer than 10 min, the resistance started to recover to baseline, even so the cells were still photo-activated. When the OptoP63 cells are activated for 2 hr, the initial drop in resistance recovers to a value above the baseline level before the activation. In addition to that, after the photo-activation period, the resistance did increase even further (*Figure 2B*, *Figure 2—figure supplement 3A*, *Figure 2—figure supplement 4A*).

When studying the entire resistance curve, OptoITSN and OptoTIAM cells run at a higher resistance than the control cells in the first 12 hr of the experiment. After the 12 hr time point, control and OptoTIAM cells run at the same median resistance where OptoITSN cells continue to run above the control cell resistance. OptoP63 cells seem to run below the resistance of control cells (*Figure 2—figure supplement 4A*). This might be indicating that there is a basal activity of the GEFs being overexpressed in the cytosol. However, the cells can clearly still be activated by recruitment of these GEFs to the plasma membrane. Furthermore, the cells can still be photo-activated after 40 hr of seeding (*Figure 2—figure supplement 4B*).

To put the resistance changes in context, it was compared to a strong activator of the barrier function, S1P. The amplitude of resistance increase induced by OptoTIAM and OptoITSN is in a similar range as the S1P-induced resistance increase. (*Figure 2—figure supplement 4C*). The photo-activation of OptoTIAM and OptoITSN did increase the resistance in a monolayer and OptoP63 activation did decrease it, all in a fast and reversible manner.

In parallel to the monolayer resistance measurements, cells were seeded for microscopy, at the same density, to image the endothelial monolayer morphology induced by photo-activation that

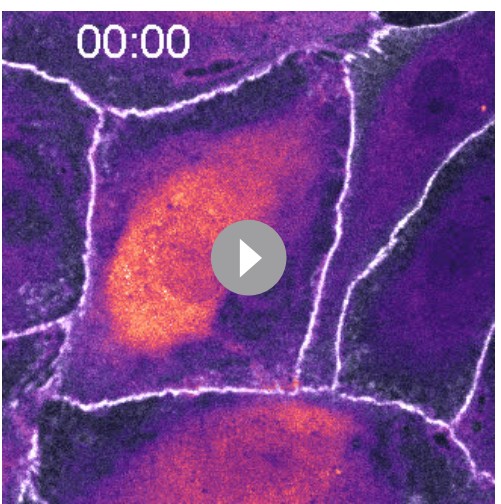

**Animation 1.** Zoom-in from a confocal microscopy time-lapse of a blood outgrowth endothelial cell (BOEC) monolayer stably expressing Lck-mTurquoise2-iLID (not shown) and SspB-HaloTag-TIAM1(DHPH) stained with JF552 nm dye (LUT = mpl-magma, bright colors indicating higher intensity). Additionally, VE-cadherin was stained with the live labeling antibody Alexa Fluor 647 Mouse Anti-Human CD144 (white). Cells were photo-activated with a 442 nm laser as indicated. Times are min:s from the start of the recording.

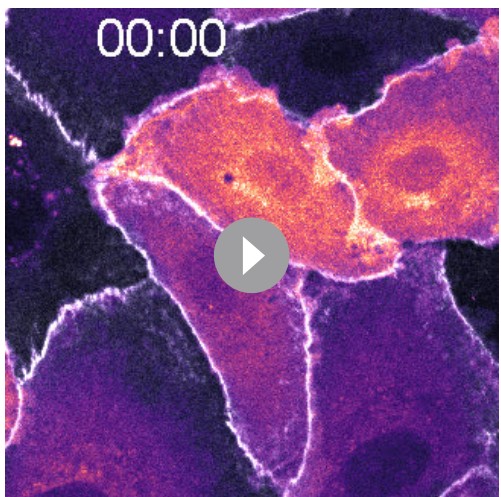

**Animation 2.** Zoom-in from a confocal microscopy time-lapse of a blood outgrowth endothelial cell (BOEC) monolayer stably expressing Lck-mTurquoise2-iLID (not shown) and SspB-HaloTag-ITSN1(DHPH) stained with JF552 nm dye (LUT = mpl-magma, bright colors indicating higher intensity). Additionally, VE-cadherin was stained with the live labeling antibody Alexa Fluor 647 Mouse Anti-Human CD144 (white). Cells were photo-activated with a 442 nm laser as indicated. Times are min:s from the start of the recording.

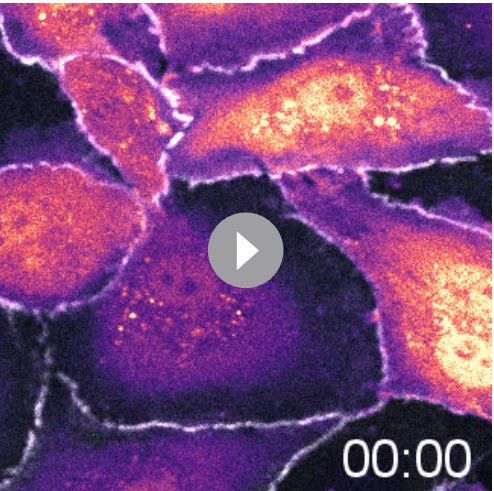

**Animation 3.** Zoom-in from a confocal microscopy time-lapse of a blood outgrowth endothelial cell (BOEC) monolayer stably expressing Lck-mTurquoise2-iLID (not shown) and SspB-HaloTag-p63RhoGEF(DH) stained with JF552 nm dye (LUT = mpl-magma, bright colors indicating higher intensity). Additionally, VE-cadherin was stained with the live labeling antibody Alexa Fluor 647 Mouse Anti-Human CD144 (white). Cells were photo-activated with a 442 nm laser as indicated. Times are min:s from the start of the recording.

would explain the rapid change in resistance. As adherens junctions are important for the barrier resistance, endothelial cells were stained for junctional marker VE-cadherin (*Figure 2C*, *Figure 2—figure supplement 2A,B*, *Figure 2—figure supplement 3B*, Animation 1–6).

For some OptoTIAM cells, increased cell-cell overlap was observed during photo-activation. Some OptoITSN cells also showed increased overlap and appeared to ruffle more during photo-activation. The adherens junctions showed a heterogenous mix of a smoother and a more jagged phenotype. During photo-activation, the balance seemed to shift toward the smoother phenotype. However, these

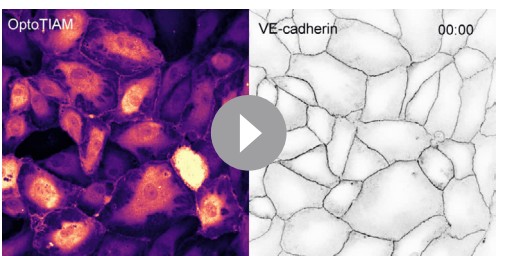

**Animation 4.** Whole field of view confocal microscopy time-lapse of a blood outgrowth endothelial cell (BOEC) monolayer stably expressing Lck-mTurquoise2-iLID (not shown) and SspB-HaloTag-TIAM1(DHPH) stained with JF552 nm dye (LUT = mpl-magma, bright colors indicating higher intensity). Additionally, VE-cadherin was stained with the live labeling antibody Alexa Fluor 647 Mouse Anti-Human CD144 (white). Cells were photo-activated with a 442 nm laser as indicated. Times are min:s from the start of the recording.

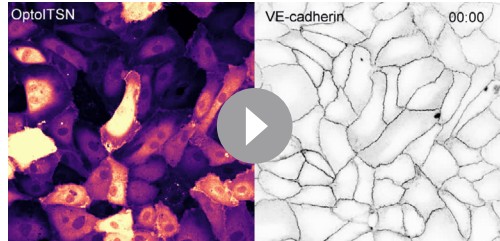

**Animation 5.** Whole field of view confocal microscopy time-lapse of a blood outgrowth endothelial cell (BOEC) monolayer stably expressing Lck-mTurquoise2-iLID (not shown) and SspB-HaloTag-ITSN1(DHPH) stained with JF552 nm dye (LUT = mpl-magma, bright colors indicating higher intensity). Additionally, VE-cadherin was stained with the live labeling antibody Alexa Fluor 647 Mouse Anti-Human CD144 (white). Cells were photo-activated with a 442 nm laser as indicated. Times are min:s from the start of the recording.

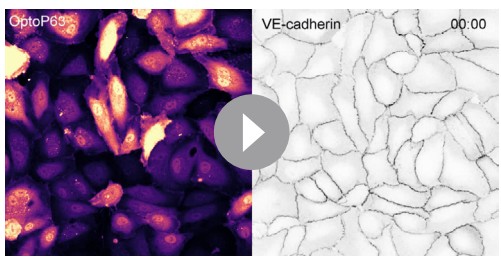

**Animation 6.** Whole field of view confocal microscopy time-lapse of a blood outgrowth endothelial cell (BOEC) monolayer stably expressing Lck-mTurquoise2-iLID (not shown) and SspB-HaloTag-p63RhoGEF(DH) stained with JF552 nm dye (LUT = mpl-magma, bright colors indicating higher intensity). Additionally, VE-cadherin was stained with the live labeling antibody Alexa Fluor 647 Mouse Anti-Human CD144 (white). Cells were photo-activated with a 442 nm laser as indicated. Times are min:s from the start of the recording.

subtle changes in phenotype could not be quantified (*Figure 2C*, *Figure 2—figure supplement 2A,B*, *Figure 2—figure supplement 3B*, Animation 1, 2, 4 and 5). OptoP63 cells clearly showed contraction, as was judged by the induction of gaps appearing in the monolayer upon photo-activation. This also changed the appearance of the VE-cadherin staining, junctions appeared more jagged, and some disappear completely, as cells detach from each other (*Figure 2C*, *Figure 2—figure supplement 2C*, *Figure 2—figure supplement 3B*, Animation 3 and 6). For the OptoP63 cells, the formation of gaps between cells caused by cell contraction and partly loss of cell-cell junctions explains the decrease in resistance and, accordingly, barrier strength. For OptoTIAM and OptoITSN it was less clear which morphological changes in the adherens junctions would explain the rapid resistance increase.

## VE-cadherin in Opto-RhoGEF induced changes in vascular barrier strength

To investigate the role of junctions in the vascular barrier strength increase/decrease induced by photo-activation of Opto-RhoGEFs, the images of VE-cadherin staining were analyzed. The linearity index is a measure of how straight the junctions are, and the straighter an endothelial junction the more mature the junction is (*Klems et al., 2020*). Values for the linearity index typically range from 1.25 to 1, where a straight junction would have a linearity index of 1 (*Otani et al., 2006*). Thus, an increased barrier strength is correlated to a linearity index value closer to 1, a decrease of the linearity index.

Comparing the junction linearity prior to activation and at the maximum of activation for Opto-TIAM, the linearity index increased slightly, for OptoITSN there was no change and for OptoP63 there was a slight increase in linearity index (*Figure 3A*, *Figure 3—figure supplement 1A*). The linearity index of OptoTIAM and OptoITSN did not explain the increased barrier strength. For OptoP63 the increased linearity index supported a role of the junctions in the decreased barrier strength upon activation. However, junctions that were dissolved during the photo-activation are not accounted for in the analysis.

Another way to describing junction properties is a temporal color-coded maximum intensity projection, displaying the junction dynamics. For OptoTIAM cells it appeared that there was a slight decrease in dynamics during the activation, in comparison to the pre-activation frames, but there appeared to be an increase in dynamics in the period right after activation (*Figure 3—figure supplement 1B*). In this recovery phase after photo-activation, there was also a dip in the resistance measurement (*Figure 2B*), which may be explained with the remodeling of the monolayer after photo-activation, including the movement of the junctions. For OptoITSN cells there is no clear change in junction dynamics upon photo-activation (*Figure 3—figure supplement 1B*). Looking at OptoP63 cells there is a clear increase in junction dynamics upon photo-activation (*Figure 3—figure supplement 1B*). It appears that higher junction dynamics occurred when resistance measurements are lower. Therefore, higher junction dynamics might explain the dip in resistance after photo-activation for OptoTIAM cells and the dip in resistance upon photo-activation for OptoP63 cells. However, there is no clear decrease in junction dynamics upon photo-activation for OptoTIAM and OptoITSN cells to explain the observed resistance increase.

It was attempted to quantify more junction properties with the 'Junction Mapper' software (*Brezovjakova et al., 2019*). However, the intensity changes over time in the time-lapse acquired in this study, with directly labeling antibody present in the medium throughout the experiment, made

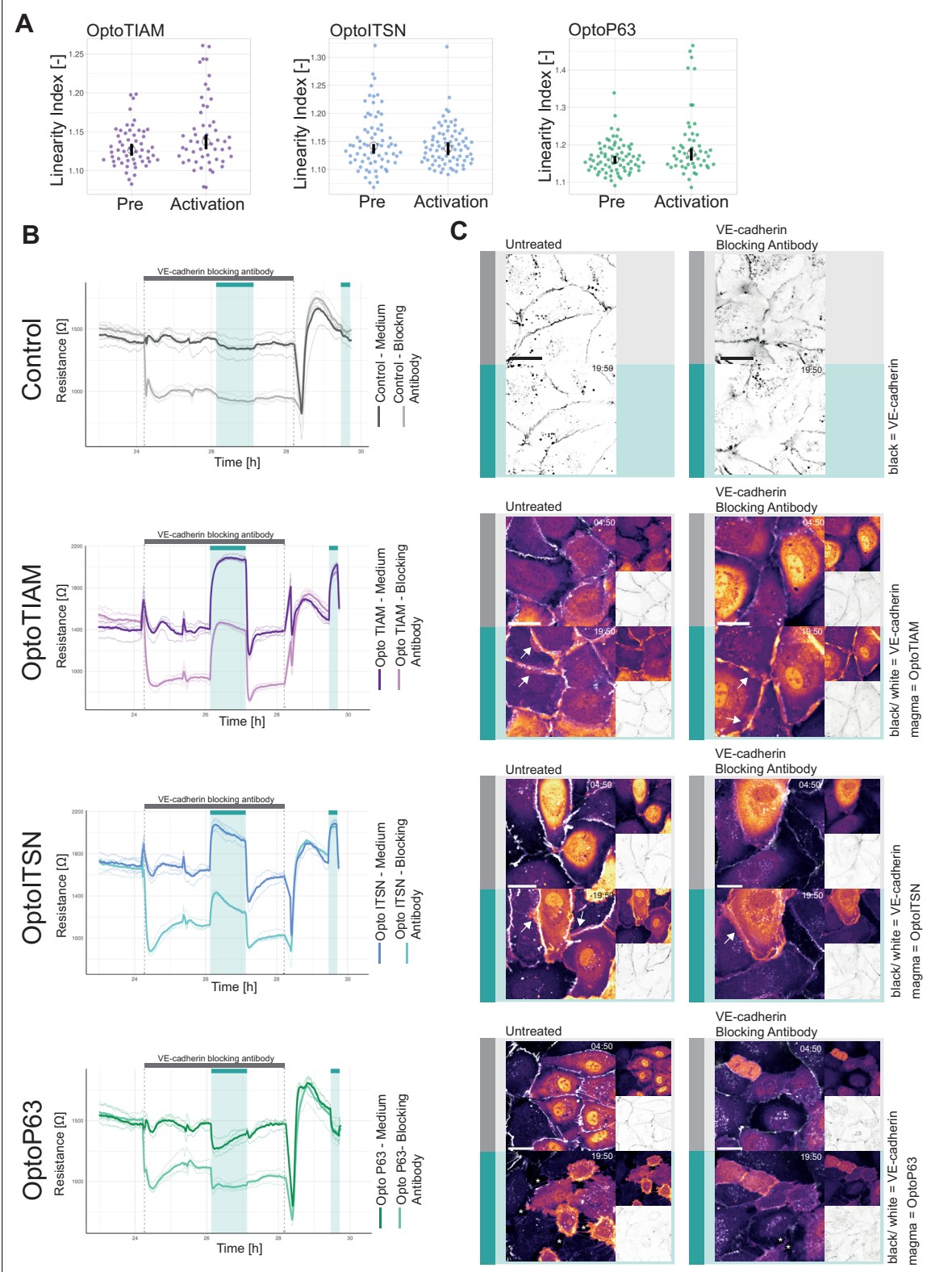

**Figure 3.** Role of junctions in optogenetically recruitable RhoGEF (Opto-RhoGEF)-induced changes in vascular barrier strength. (**A**) Linearity index for a blood outgrowth endothelial cell (BOEC) monolayer stably expressing Lck-mTurquoise2-iLID and either SspB-HaloTag-TIAM1(DHPH) (purple)/ ITSN1(DHPH) (blue) or p63RhoGEF(DH) (green) for the frame before photo-activation (pre) and the last frame of photo-activation (Activation). Images of the junctions are in *Figure 2—figure supplement 2* and example of linearity index analysis in *Figure 3—figure supplement 1A*. Each

*Figure 3 continued on next page*

*Figure 3 continued*

dot represents an individual cell. The median of the data is shown as a black circle and the 95% confidence interval for each median, determined by bootstrapping, is indicated by the bar. The number of cells is: OptoTIAM pre = 55, OptoTIAM Activation = 52, OptoITSN pre = 71, OptoITSN Activation = 76, OptoP63 pre = 74, OptoP63 Activation = 51. The data is from two independent experiments. (**B**) Resistance of a monolayer of BOECs stably expressing Lck-mTurquoise2-iLID, solely as a control (gray), and either SspB-HaloTag-TIAM1(DHPH)(purple)/ITSN1(DHPH) (blue) or p63RhoGEF(DH) (green) measured with electrical cell-substrate impedance sensing (ECIS) at 4000 Hz, representing paracellular permeability, every 10 s. Cyan bars indicated photo-activation with blue LED light (60 min, 15 min). Gray bar with dashed lines indicates the addition of VE-cadherin blocking antibody in medium (darker color line) or medium as a control (lighter color line). At the end of the gray bar the medium is replaced for all conditions. Thin lines represent the average value from one well of an 8W10E PET ECIS array. Thick lines represent the mean. Four wells were measured for each condition. (**C**) Representative zoom-ins from confocal microscopy images of a BOEC monolayer stably expressing Lck-mTurquoise2-iLID (not shown) and either SspB-HaloTag-TIAM1(DHPH)/ITSN1(DHPH) or p63RhoGEF(DH) stained with JF552 nm dye (LUT = mpl-magma, bright colors indicating higher intensity). Additionally, VE-cadherin was stained with the live labeling antibody Alexa Fluor 647 Mouse Anti-Human CD144 (white in merge, gray inverted in single channel). Left panel shows untreated cells, right panel shows cells treated with the VE-cadherin blocking antibody. Scale bars: 25 µm. Times are min:s from the start of the recording. Gray bar indicates the condition before photo-activation. Cyan bar indicates 442 nm photo-activation. Arrows indicate overlap and protrusions. Asterisks indicate holes in monolayer.

The online version of this article includes the following figure supplement(s) for figure 3:

**Figure supplement 1.** Junction analysis.

**Figure supplement 2.** Replicate electrical cell-substrate impedance sensing (ECIS) assay in the presence of the VE-cadherin blocking antibody.

intensity-based measurements incomparable for different time points. Therefore, these measurements were not informative and not presented here.

To further explore which role the junctions play in the vascular barrier strength increase triggered by GEFs, it was investigated how the monolayer responded to Opto-RhoGEF activation in the absence of VE-cadherin junctions. Therefore, a VE-cadherin blocking antibody was used. Opto-RhoGEFs and control cells were grown into a monolayer on an ECIS array and treated with VE-cadherin blocking antibody. The VE-cadherin blocking antibody caused the homodimeric bonds to break between the VE-cadherins (*Figure 3—figure supplement 1C*, Animation 7). Previous work has shown the specific blocking effect of this antibody in comparison to the VE-cadherin (55-7H1) labeling antibody (*Kroon et al., 2014*). The VE-cadherin blocking antibody lowered the resistance in all conditions, which dropped to a stable level (*Figure 3B*, *Figure 3—figure supplement 2A*). Subsequently, the cells were photo-activated and for OptoTIAM and OptoITSN, an increase in resistance was measured comparable to the control conditions without the VE-cadherin blocking antibody. The amplitude of resistance appeared to be the same. OptoP63 showed a decrease in resistance, but the drop is not more severe in the presence of VE-cadherin blocking antibody. The VE-cadherin blocking antibody was washed out and the resistance returned to baseline matching the control condition. Subsequently, both conditions were photo-activated and showed a response that was similar between the two conditions. This suggests that the entire system is reversible with respect to the antibody treatment and optogenetic manipulation. In parallel to the ECIS experiment, cells were seeded for microscopy at the same density and treated with the VE-cadherin blocking antibody. The junctions looked more diffuse in the treated conditions (*Figure 3C*, Animation 8–15). For OptoTIAM and OptoITSN cells there were no striking differences in the appearance of VE-cadherin stain upon photo-activation in both conditions, but both showed membranes that

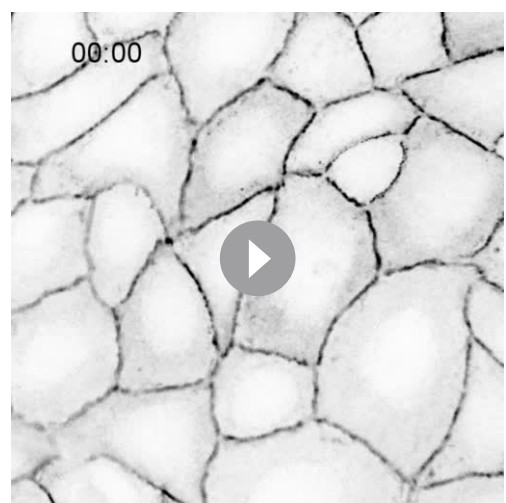

**Animation 7.** Confocal microscopy time-lapse of a blood outgrowth endothelial cell (BOEC) monolayer stably expressing Lck-mTurquoise2-iLID (not shown) and SspB-HaloTag-TIAM1(DHPH) (not shown), additionally, VE-cadherin was stained with the live labeling antibody Alexa Fluor 647 Mouse Anti-Human CD144 (gray inverted). Cells were treated with VE-cadherin blocking antibody as indicated. Times are min:s from the start of the recording.

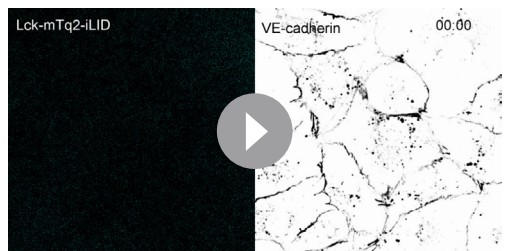

**Animation 8.** Whole field of view confocal microscopy time-lapse of a blood outgrowth endothelial cell (BOEC) monolayer stably expressing Lck-mTurquoise2-iLID (cyan). Additionally, VE-cadherin was stained with the live labeling antibody Alexa Fluor 647 Mouse Anti-Human CD144 (gray inverted). Cells were photo-activated with a 442 nm laser as indicated. Times are min:s from the start of the recording.

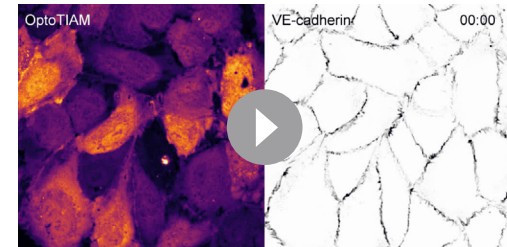

**Animation 10.** Whole field of view confocal microscopy time-lapse of a blood outgrowth endothelial cell (BOEC) monolayer stably expressing Lck-mTurquoise2-iLID (not shown) and SspB-HaloTag-TIAM1(DHPH) stained with JF552 nm dye (LUT = mpl-magma, bright colors indicating higher intensity). Additionally, VE-cadherin was stained with the live labeling antibody Alexa Fluor 647 Mouse Anti-Human CD144 (gray inverted). Cells were photo-activated with a 442 nm laser as indicated. Times are min:s from the start of the recording.

overlapped (*Figure 3C*, Animation 10–13). For OptoP63 cells, VE-cadherin showed junction disassembly and gaps in the monolayer upon photo-activation in both conditions (*Figure 3C*, Animation 14 and 15).

Under conditions where blocking of VE-cadherin reduced the resistance of endothelial monolayers, the barrier could still be enhanced, independent from VE-cadherin. By photo-activating specific GEFs that subsequently activate Rho GTPases Rac1 or Cdc42, the endothelial barrier function can be increased.

## Morphology changes induced by global activation of Opto-RhoGEFs in subconfluent endothelial cells

To identify the mechanism that allowed for barrier strength increase in the absence of VE-cadherin, it was attempted to quantify cell-cell overlap in the monolayer. The attempt to stain two populations of Opto-RhoGEF cells with different HaloTag JF dyes to then grow them in a mosaic monolayer, to measure membrane overlap at junction regions directly, failed. The two populations could not be distinguished by fluorescence color anymore after the 24 hr that it takes for the cells to form a monolayer. In images of Opto-RhoGEF stained in one

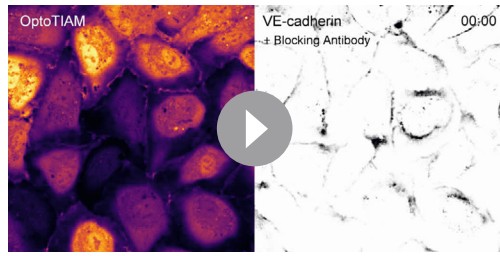

**Animation 11.** Whole field of view confocal microscopy time-lapse of a blood outgrowth endothelial cell (BOEC) monolayer stably expressing Lck-mTurquoise2-iLID (not shown) and SspB-HaloTag-TIAM1(DHPH) stained with JF552 nm dye (LUT = mpl-magma, bright colors indicating higher intensity) and treated with the VE-cadherin blocking antibody. Additionally, VE-cadherin was stained with the live labeling antibody Alexa Fluor 647 Mouse Anti-Human CD144 (gray inverted). Cells were photo-activated with a 442 nm laser as indicated. Times are min:s from the start of the recording.

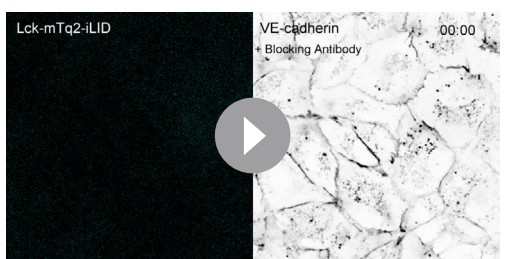

**Animation 9.** Whole field of view confocal microscopy time-lapse of a blood outgrowth endothelial cell (BOEC) monolayer stably expressing Lck-mTurquoise2-iLID (cyan) and treated with the VE-cadherin blocking antibody. Additionally, VE-cadherin was stained with the live labeling antibody Alexa Fluor 647 Mouse Anti-Human CD144 (gray inverted). Cells were photo-activated with a 442 nm laser as indicated. Times are min:s from the start of the recording.

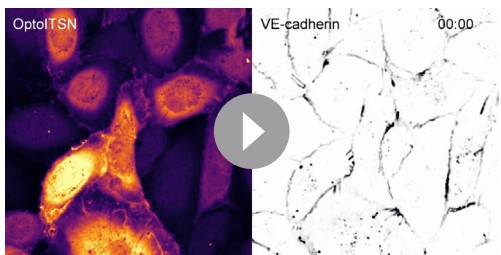

**Animation 12.** Whole field of view confocal microscopy time-lapse of a blood outgrowth endothelial cell (BOEC) monolayer stably expressing Lck-mTurquoise2-iLID (not shown) and SspB-HaloTag-ITSN1(DHPH) stained with JF552 nm dye (LUT = mpl-magma, bright colors indicating higher intensity). Additionally, VE-cadherin was stained with the live labeling antibody Alexa Fluor 647 Mouse Anti-Human CD144 (gray inverted). Cells were photo-activated with a 442 nm laser as indicated. Times are min:s from the start of the recording.

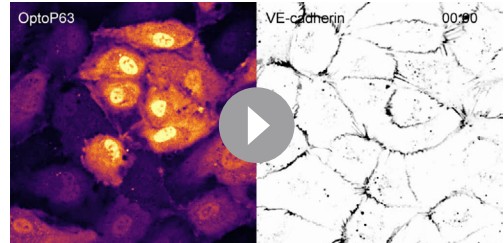

**Animation 14.** Whole field of view confocal microscopy time-lapse of a blood outgrowth endothelial cell (BOEC) monolayer stably expressing Lck-mTurquoise2-iLID (not shown) and SspB-HaloTag-p63RhoGEF(DH) stained with JF552 nm dye (LUT = mpl-magma, bright colors indicating higher intensity). Additionally, VE-cadherin was stained with the live labeling antibody Alexa Fluor 647 Mouse Anti-Human CD144 (gray inverted). Cells were photo-activated with a 442 nm laser as indicated. Times are min:s from the start of the recording.

color cell-cell overlap area could not be measured in a reliable manner. Therefore, the cell area change was studied in subconfluent endothelial cells.

The initial cell area was measured before photo-activation to rule out that expressing the Opto-RhoGEFs in the cytosol by itself had influence on the cell size (*Figure 4—figure supplement 1A*). To assess the effect of optogenetics activation on cell morphology, cells were illuminated with 442 nm light for 10 min and the cell area before activation and at the peak of extension/spreading were super-imposed (*Figure 4A*). Control cells only expressing Lck-mTurquoise2-iLID showed extension on one side of the cell and contraction at the opposite side, they appeared to move in one direction.

OptoTIAM and OptoITSN cells spread out in all directions upon activation. Therefore, these cells appeared rounder. This was quantified by calculating the form factor (form factor 1 is a perfect circle), which confirmed the observation (*Figure 4—figure supplement 1C*). In contrast, OptoP63 cells contracted and were less round (*Figure 4A*, *Figure 4—figure supplement 1C*).

Following the cell area over time during photo-activation showed that OptoTIAM and OptoITSN increased the cell area (*Figure 4B*). Cells expressing OptoTIAM increased about 15% in size and reached the maximum at 10 min photo-activation. In comparison, OptoITSN expressing cells reached

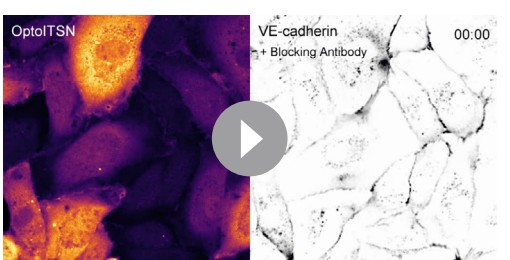

**Animation 13.** Whole field of view confocal microscopy time-lapse of a blood outgrowth endothelial cell (BOEC) monolayer stably expressing Lck-mTurquoise2-iLID (not shown) and SspB-HaloTag-ITSN1(DHPH) stained with JF552 nm dye (LUT = mpl-magma, bright colors indicating higher intensity) and treated with the VE-cadherin blocking antibody. Additionally, VE-cadherin was stained with the live labeling antibody Alexa Fluor 647 Mouse Anti-Human CD144 (gray inverted). Cells were photo-activated with a 442 nm laser as indicated. Times are min:s from the start of the recording.

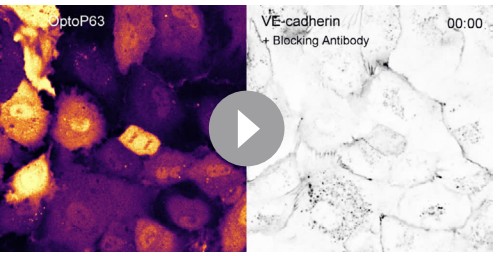

**Animation 15.** Whole field of view confocal microscopy time-lapse of a blood outgrowth endothelial cell (BOEC) monolayer stably expressing Lck-mTurquoise2-iLID (not shown) and SspB-HaloTag-p63RhoGEF(DH) stained with JF552 nm dye (LUT = mpl-magma, bright colors indicating higher intensity) and treated with the VE-cadherin blocking antibody. Additionally, VE-cadherin was stained with the live labeling antibody Alexa Fluor 647 Mouse Anti-Human CD144 (gray inverted). Cells were photo-activated with a 442 nm laser as indicated. Times are min:s from the start of the recording.

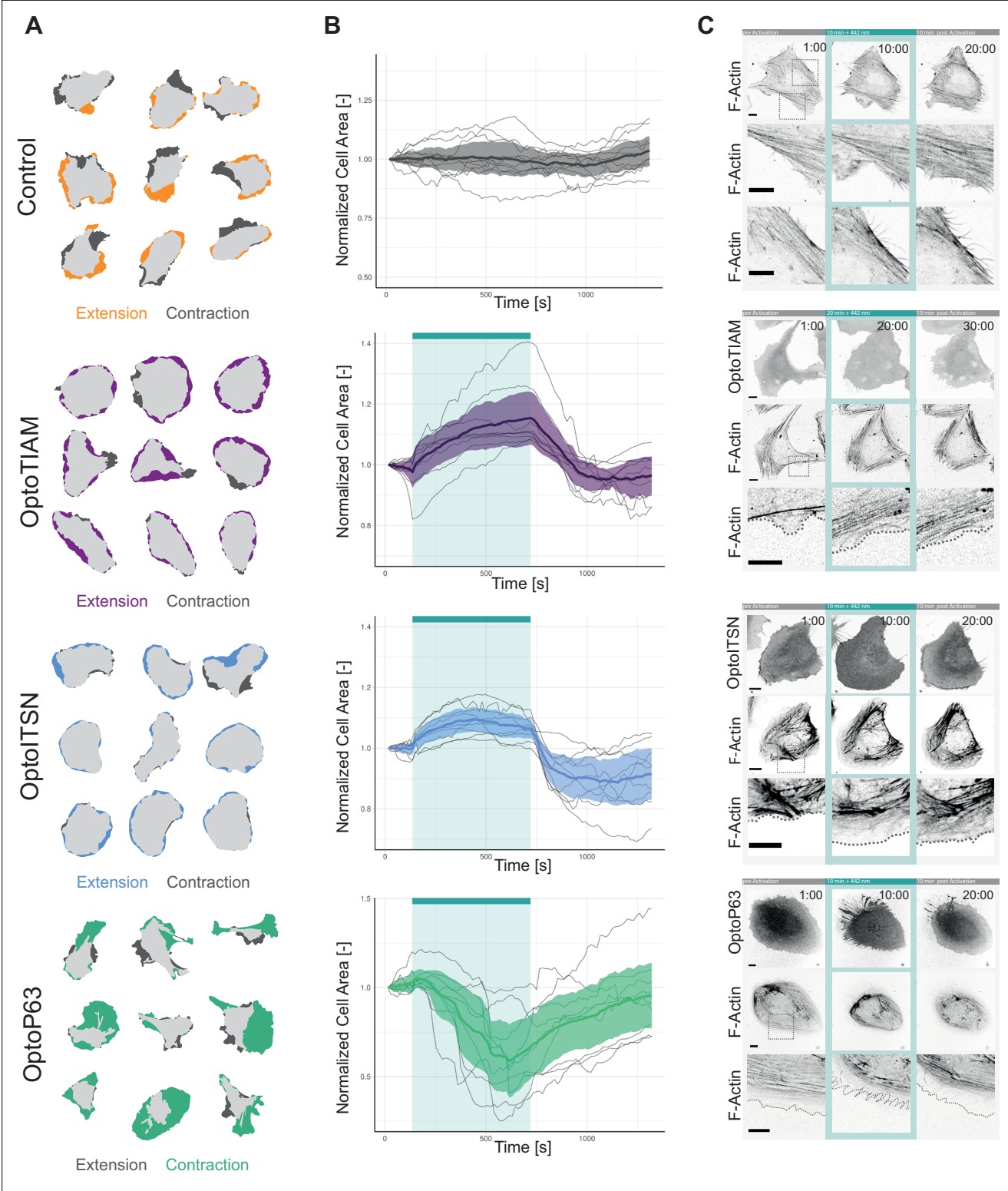

**Figure 4.** Characterization of global photo-activation of blood outgrowth endothelial cells (BOECs) expressing optogenetically recruitable RhoGEFs (Opto-RhoGEFs). (**A**) Cell area change during 10 min photo-activation with a 442 nm laser line for BOECs stably expressing Lck-mTurquoise2-iLID, solely as control, and either SspB-HaloTag-TIAM1(DHPH)/ ITSN1(DHPH) or p63RhoGEF(DH). The images show an overlay of the cell area before and after the photo-activation and colors indicate either contraction or extension. (**B**) Normalized cell area over time for the cells depicted in **A**. Control = gray, OptoTIAM=purple, OptoITSN=blue, OptoP63=green. The cyan bar indicates photo-activation with 442 nm laser light. Thin lines represent individual cells, thick lines represent the mean values, and ribbons represent their 95% confidence interval. The number of analyzed cells is: Control

*Figure 4 continued on next page*

*Figure 4 continued*

= 9, OptoTIAM=9, OptoITSN=9, OptoP63=9. The data is from two independent experiments and at least three independent photo-activations. (**C**) Representative confocal microscopy images of BOECs stably expressing Lck-mTurquoise2-iLID (not shown), solely as control, and either SspB-HaloTag-TIAM1(DHPH)/ITSN1(DHPH) or p63RhoGEF(DH) stained with JF552 nm dye. Additionally, stained for F-actin with SiR-actin. Scale bars: 10 µm. Times are min:s from the start of the recording. Cyan bar indicates 442 nm photo-activation. Gray box indicates zoom. Gray dashed line indicates cell edge.

The online version of this article includes the following figure supplement(s) for figure 4:

**Figure supplement 1.** Global photo-activation.

**Figure supplement 2.** Cell-cell overlap.

---

the cell area maximum of 10% after roughly 3 min and after that the cell area slightly decreased during the photo-activation. After photo-activation both OptoTIAM and OptoITSN expressing cells decreased rapidly in cell area below the baseline (*Figure 4B*). The opposite effect was observed for OptoP63 expressing cells, they started decreasing in cell area after about 1 min of photo-activation and reached a minimum of roughly 40% cell area decrease at 7 min. After this time point the OptoP63 expressing cells already recovered and increased in size, even though they were still photo-activated (*Figure 4B*). Following the cell area of control cells over time showed fluctuation in cell area but on average they stayed the same size (*Figure 4B*). A larger pool of cells was analyzed for three time points, confirming the findings of the cell size over time measurements (*Figure 4—figure supplement 1B*). In conclusion, photo-activation of Opto-RhoGEFs controls cell size to a certain extent. A study using the rapamycin system to recruit the DHPH domain of TIAM1 and the DH domain of p63RhoGEF in endothelial cells found similar changes in cell area, about 15% increase for TIAM1 recruitment and about 25% decrease for p63RhoGEF recruitment (*Reinhard et al., 2017*). Even though the rapamycin showed a higher recruitment efficiency the iLID system seems able to induce the maximal cellular response.

Actin fibers define the morphology of the cell and actin polymerization is the driving force of cellular movement (*Noda et al., 2010*), therefore the Opto-RhoGEF cells were stained for F-actin to study changes in actin structures upon photo-activation (*Figure 4C*, Animation 16–19). For the control, actin fibers are visible at the cell edge, in an orientation parallel to the edge. In the OptoTIAM cells, the actin fibers parallel to the cell edge seemed to disassemble during activation and areas of cell expansions showed no thick actin fibers. Areas with actin fibers perpendicular to the cell edge rather showed slight contraction instead of cell spreading. Also, OptoITSN cells showed less dense actin structures in the extending areas of the activated cell. OptoP63 cells showed actin fibers clustering together during cell contraction upon photo-activation. During recovery, the new cell edges also showed very little actin staining. It needs to be noted that we used SiR-actin to stain the actin cytoskeleton. SiR-actin supposedly stains more mature actin structures, which might underrepresent the newly polymerized actin structures (*Simao et al., 2021*).

To investigate if the cell size, specifically cell-cell overlap in a monolayer, plays a role in the observed barrier strength increase, the time scale of changes in cell area and resistance were compared. To this end, the cell area change of subconfluent cells, upon activation (*Figure 4B*), and the vascular barrier strength upon activation (*Figure 2B*) were compared and showed a correlation (*Figure 4—figure supplement 2A and B*). This supports the idea that the increased

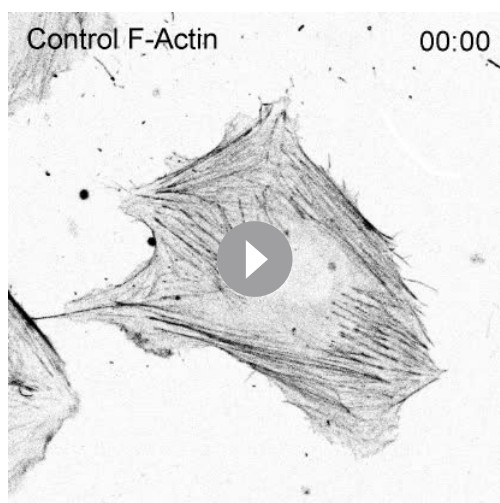

**Animation 16.** Confocal microscopy time-lapse of a blood outgrowth endothelial cell (BOEC) stably expressing Lck-mTurquoise2-iLID (not shown), solely as control. Additionally, stained for F-actin with SiR-actin. Cell was photo-activated with a 442 nm laser line as indicated. Times are min:s from the start of the recording.

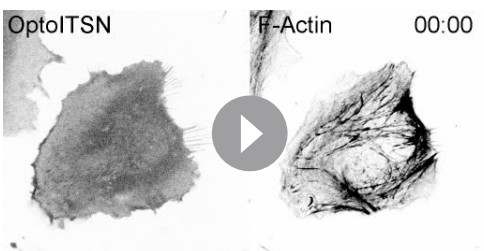

**Animation 17.** Confocal microscopy time-lapse of a blood outgrowth endothelial cell (BOEC) stably expressing Lck-mTurquoise2-iLID (not shown) and SspB-HaloTag-TIAM1(DHPH) stained with JF552 nm dye (left). Additionally, stained for F-actin with SiR-actin (right). Cell was photo-activated with a 442 nm laser line as indicated. Times are min:s from the start of the recording.

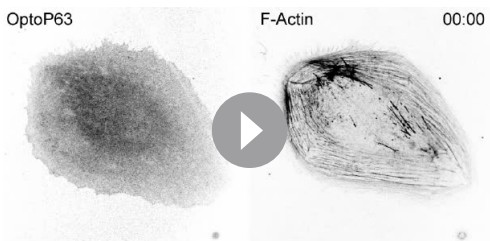

**Animation 19.** Confocal microscopy time-lapse of a blood outgrowth endothelial cell (BOEC) stably expressing Lck-mTurquoise2-iLID (not shown) and SspB-HaloTag-p63RhoGEF(DH) stained with JF552 nm dye (left). Additionally, stained for F-actin with SiR-actin (right). Cell was photo-activated with a 442 nm laser line as indicated. Times are min:s from the start of the recording.

endothelial membrane overlaps at junction regions by itself resulted in increased resistance, assuming that cell spreading in a monolayer increases overlap.

To further investigate this, OptoTIAM cells in a monolayer were stained with PECAM1, an endothelial cell-specific adhesion molecule that has been observed in overlapping protrusions. We indeed observed PECAM1 prominently present at endothelial membrane overlaps. Upon activation, the PECAM1-positive area increased and colocalized with the areas with increased intensity in the OptoTIAM channel (*Figure 4—figure supplement 2C*, Animation 20). Thus, the OptoTIAM channel could also be used to make an estimate of overlap, as the PECAM1 intensity increased where two cells overlap. However, this only works if the cells have roughly the same intensity. The intensity range was too heterogeneous in this dataset to analyze the entire monolayer in the field of view, but example images are shown. Within the first 2 min of activation, the time it takes for the barrier strength to increase at least the halftime point, the OptoTIAM cells showed increased overlap but no clear change in VE-cadherin staining (*Figure 4—figure supplement 2D, E, and F*; Animation 4). Studying the subconfluent monolayer, the activation of OptoTIAM increased the area covered by the cells, through the closure of gaps in the monolayer (*Figure 4—figure supplement 2G*). The cells covered the gaps first with membrane, there was a clear cell extension after 2 min but no new junctions are formed yet (*Figure 4—figure supplement 2H*). After 15 min, the first new junctions started to appear and after 30 min these junctions showed a linear phenotype (*Figure 4—figure supplement 2H*, Animation 21). Additionally, cell-cell membrane overlap increased about 20%, up on photo-activation of

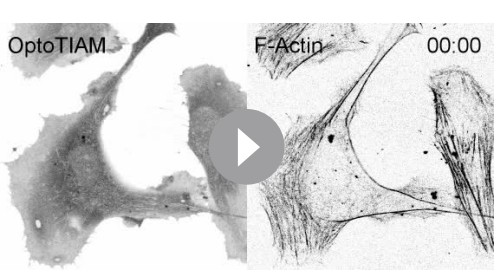

**Animation 18.** Confocal microscopy time-lapse of a blood outgrowth endothelial cell (BOEC) stably expressing Lck-mTurquoise2-iLID (not shown) and SspB-HaloTag-ITSN1(DHPH) stained with JF552 nm dye (left). Additionally, stained for F-actin with SiR-actin (right). Cell was photo-activated with a 442 nm laser line as indicated. Times are min:s from the start of the recording.

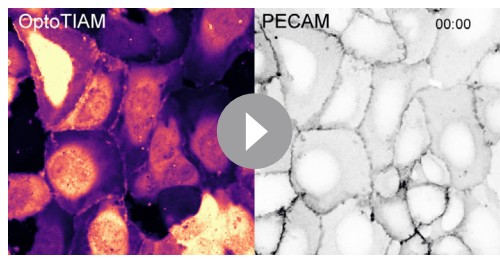

**Animation 20.** Confocal time-lapse of a blood outgrowth endothelial cell (BOEC) monolayer stably expressing Lck-mTurquoise2-iLID (not shown) and SspB-HaloTag-TIAM1(DHPH) (left, LUT = mpl-magma, brighter colors represent higher intensity), additionally, PECAM was stained with a live labeling antibody Alexa Fluor 647 Mouse Anti-Human CD31 (right, inverted gray). Cells were photo-activated with 442 nm laser light as indicated. Times are min:s from the start of the recording.

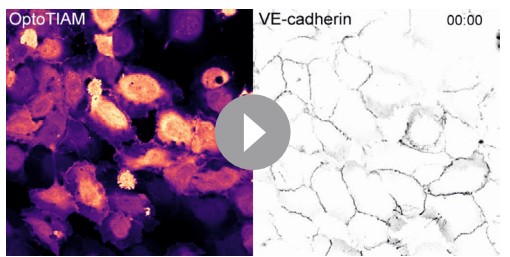

**Animation 21.** Whole field of view confocal microscopy time-lapse of blood outgrowth endothelial cells (BOECs), growing in a subconfluent monolayer, stably expressing Lck-mTurquoise2-iLID (not shown) and SspB-HaloTag-TIAM1(DHPH) (left, LUT = mpl-magma, brighter colors represent higher intensity), additionally, VE-cadherin was stained with the live labeling antibody Alexa Fluor 647 Mouse Anti-Human CD144 (right, inverted gray). Cells were photo-activated with 442 nm laser light as indicated. Time is min:s from the start of the recording.

OptoTIAM, in a mosaic expression monolayer (*Figure 4—figure supplement 2I and J*, Animation 22).

In conclusion, endothelial barrier function is not solely determined by the junctional protein VE-cadherin. Clearly, overlapping membranes at junction regions can also promote endothelial barrier. The overlapping membranes are potentially induced by increased cell size of single endothelial cells that once they are present in a monolayer, consequently partly overlap.

## Local activation of Opto-RhoGEFs in subconfluent endothelial cells

Rho GTPase activity is defined on a subcellular level and not homogenously over the entire cell (*Rossman et al., 2005*). Therefore, the Opto-RhoGEFs were additionally tested for their potential to locally induce Rho GTPase activity and thereby morphology changes on a subcellular scale. In addition, we studied the plasticity of the endothelial cells by photo-activating different spots.

Hence, cells were locally photo-activated with 442 nm laser light for 10 min in the indicted region of interest (ROI). After 10 min, this activation ROI was moved to the opposite side of the cell. OptoTIAM cells and OptoITSN cells extended rather precisely in the ROI, as indicated by blue and purple colors (*Figure 5A*, Animation 23 and 24). OptoP63 cells contracted and moved out of the ROI, as indicated in green (*Figure 5A*, Animation 25). Following multiple cells, OptoTIAM cells increased coverage of the activation ROI after 10 min by about 20% (*Figure 5B*). OptoITSN cells increased the ROI coverage by 30% within 10 min of activation. OptoP63 cells decreased the ROI coverage for about 40% after 10 min of activation. Measuring ROIs for unstimulated control cells, no average change in coverage was measured (*Figure 5—figure supplement 1A*). Cells were sequentially activated at two different positions, and the response was equally strong for the second activation, indicating no immediate fatigue of the system (*Figure 5—figure supplement 1B*).

By studying the actin structures to define cell morphology, we observed morphological changes at the activation ROIs. Less thick actin bundles were detected in the activated area for OptoTIAM and OptoITSN (*Figure 5C*, Animation 26 and 27). OptoP63 showed more actin fibers in the activated ROI (*Figure 5C*, Animation 28). Additionally, it was possible to drive an OptoTIAM cell toward an ROI. First, the cell extended into the ROI, and by moving the ROI, the entire cell eventually followed the direction of the ROI (*Figure 5—figure supplement 1C*, Animation 29 and 30). This showed the strong potential of Opto-RhoGEFs to drive directed cell migration and induce local cell protrusions rapidly and reversibly.

## Discussion

The endothelium forms the barrier between blood and tissue, and it is known that Rho GTPases play a role in the barrier maintenance but how it is exactly regulated by the temporal and spatial defined Rho GTPase activity remains unknown. The vascular barrier integrity is challenged on a subcellular level by, for example, transmigration of leukocytes but also on a global level, for example during chronic inflammation. Here, we

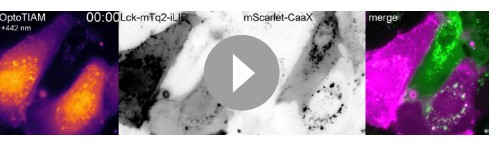

**Animation 22.** Wide-field microscopy time-lapse of a mosaic blood outgrowth endothelial cell (BOEC) monolayer with one population stably expressing SspB-HaloTag-TIAM1(DHPH) (first panel, LUT = mpl-magma, brighter colors represent higher intensity) and Lck-mTurquoise2-iLID (second panel), and the other population stably expressing the membrane marker mScarlet-CaaX (third panel). The cell-cell overlap is represented by the merge (forth panel) of the Lck-mTurquoise2-iLID (magenta) and mScarlet-CaaX (green) channel. Cells were photo-activated with 442 nm excitation light as indicated. Time is min:s from the start of the recording.

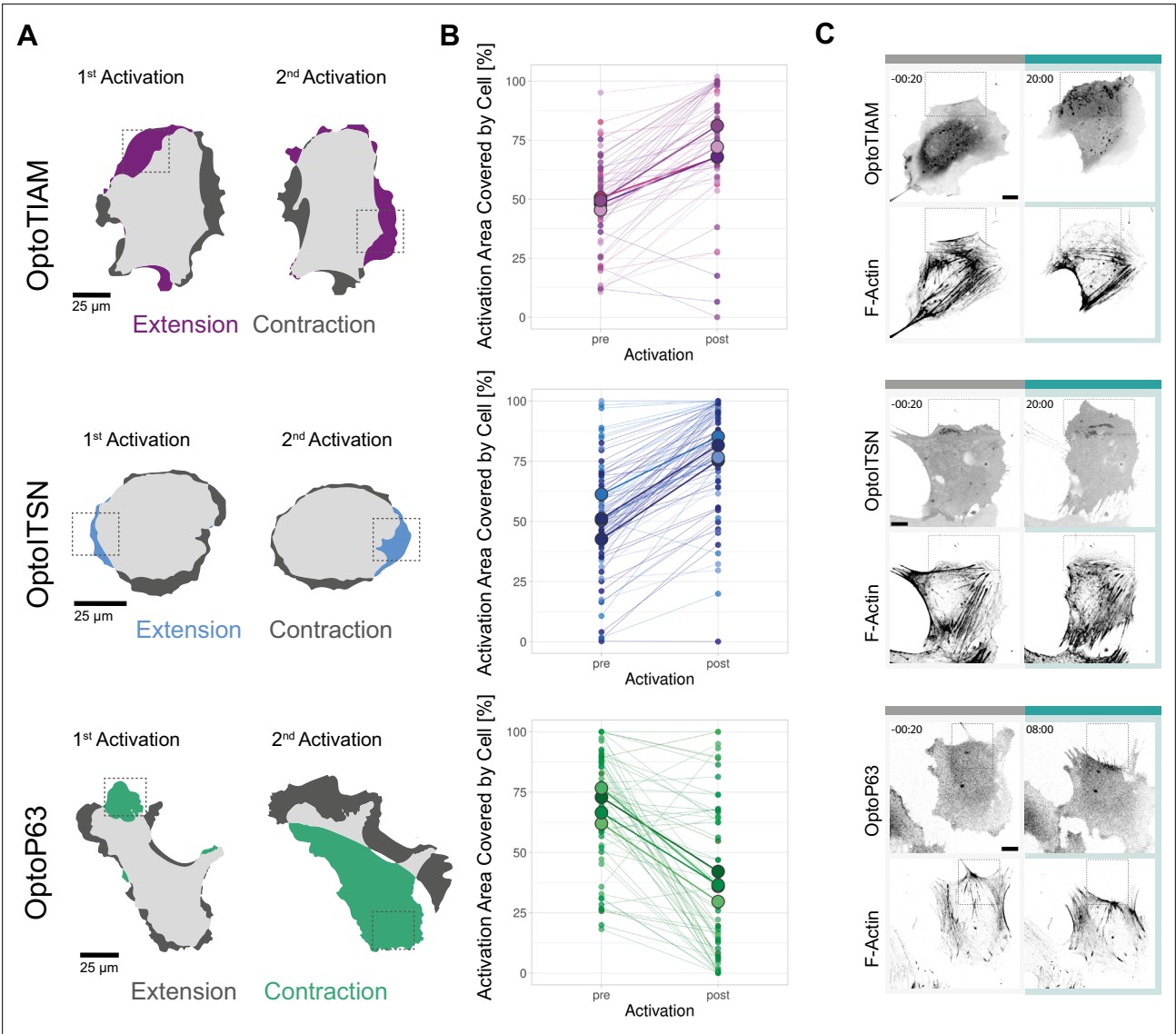

**Figure 5.** Characterization of local photo-activation of blood outgrowth endothelial cells (BOECs) expressing optogenetically recruitable RhoGEFs (Opto-RhoGEFs). (**A**) Cell area change during two 10 min local photo-activations with 442 nm laser line for BOECs stably expressing Lck-mTurquoise2-iLID and either SspB-HaloTag-TIAM1(DHPH)/ITSN1(DHPH) or p63RhoGEF(DH). The images show an overlay of the cell area before and after the activation and colors indicate either contraction or extension. The gray dashed box indicates the area of activation. (**B**) Activation area covered by cell, pre- and post-10 min activation with 442 nm laser line, for BOECs stably expressing Lck-mTurquoise2-iLID and either SspB-HaloTag-TIAM1(DHPH) (purple)/ITSN1(DHPH) (blue) or p63RhoGEF(DH) (green). Small dots represent individual activation areas, which are connected by lines. Larger dots represent the mean of each replicate indicated by different colors. The number of activation areas for OptoTIAM=53, for OptoITSN=85, OptoP63=61. (**C**) Representative confocal microscopy images of BOECs stably expressing Lck-mTurquoise2-iLID (not shown) and either SspB-HaloTag-TIAM1(DHPH)/ITSN1(DHPH) or p63RhoGEF(DH) stained with JF552 nm dye. Additionally, stained for F-actin with SiR-actin. Scale bars: 10 µm. Times are min:s from the start of the photo-activation. Cyan bar indicates 442 nm photo-activation and gray box indicates area of activation.

The online version of this article includes the following figure supplement(s) for figure 5:

**Figure supplement 1.** Local photo-activation.

apply an optogenetic tool with the unique ability to reversibly induce spatially and temporally defined Rho GTPase activity to explore its potential in studying vascular barrier strength and morphology of endothelial cells. The iLID optogenetic heterodimerization tool was chosen to recruit catalytic active domains of GEFs, the activators of Rho GTPases, to the plasma membrane, where they induce activity of specific Rho GTPases. Endothelial cells lines were generated expressing Lck-mTurquoise2-iLID and either SspB-HaloTag-TIAM1(DHPH), -ITSN1(DHPH), or -p63RhoGEF(DH) to activate Rac, Cdc42, and

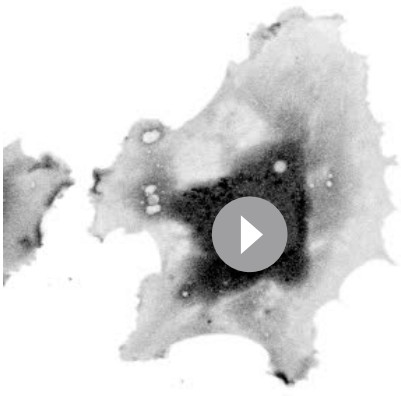

**Animation 23.** Confocal time-lapse of a blood outgrowth endothelial cell (BOEC) stably expressing Lck-mTurquoise2-iLID (not shown) and SspB-HaloTag-TIAM1(DHPH) (gray inverted) stained with JF552 nm dye, two times locally photo-activated for 10 min with a 442 nm laser, as indicated by the cyan dashed box. Times are min:s from the start of the recording.

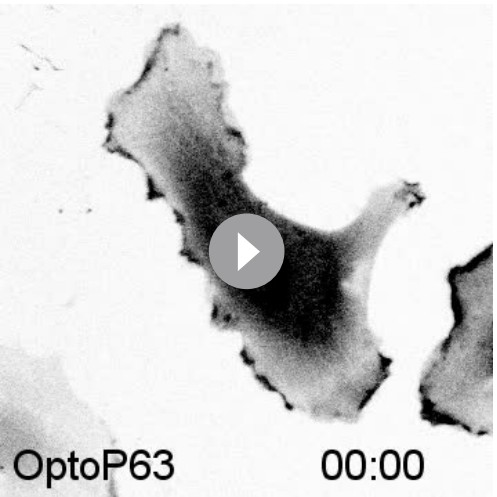

**Animation 25.** Confocal time-lapse of a blood outgrowth endothelial cell (BOEC) stably expressing Lck-mTurquoise2-iLID (not shown) and SspB-HaloTag-p63RhoGEF(DH) (gray inverted), stained with JF552 nm dye, two times locally photo-activated for 10 min with a 442 nm laser, as indicated by the cyan dashed box. Times are min:s from the start of the recording.

Rho respectively. Resistance measurements in monolayers revealed precisely inducible and reversible increase in vascular barrier strength for Rac and Cdc42 activation and a decrease in Rho activation, illustrating the temporal control of the optogenetics tool. To the best of our knowledge this is the first time optogenetic control over vascular barrier strength is being reported. To further study the Rho GTPase, induced cellular processes resulting in changed vascular barrier strength, microscopy, and a VE-cadherin blocking antibody experiment were performed. Surprisingly, while the main protein of adherens junction, that is VE-cadherin homotypic interactions, was blocked, the vascular barrier strength could still be changed by the optogenetic recruitment of GEFs. Additionally, the microscopy experiments suggested a contribution of cell-cell membrane overlap to the vascular barrier strength, while no clear changes in junction appearance were observed on the same time scale. Furthermore, we demonstrate whole cell and subcellular control over endothelial cell

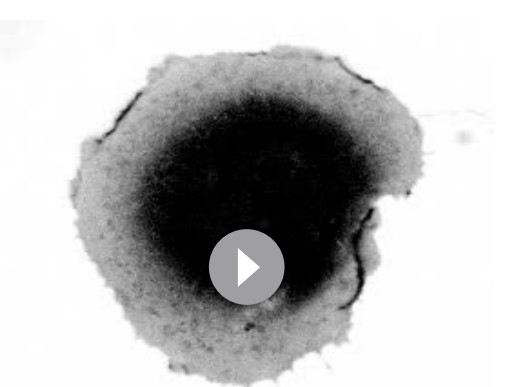

**Animation 24.** Confocal time-lapse of a blood outgrowth endothelial cell (BOEC) stably expressing Lck-mTurquoise2-iLID (not shown) and SspB-HaloTag-ITSN1(DHPH) (gray inverted), stained with JF552 nm dye, two times locally photo-activated for 10 min with a 442 nm laser, as indicated by the cyan dashed box. Times are min:s from the start of the recording.

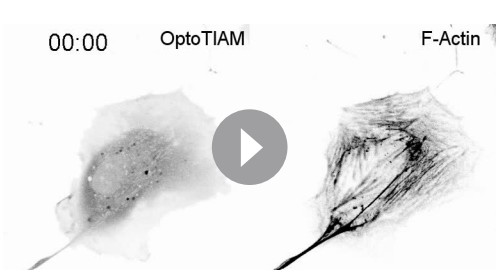

**Animation 26.** Confocal time-lapse of a blood outgrowth endothelial cell (BOEC) stably expressing Lck-mTurquoise2-iLID (not shown) and SspB-HaloTag-TIAM1(DHPH) (left) stained with JF552 nm dye. Additionally, stained for F-actin with SiR-actin (right). The cell was locally photo-activated with a 442 nm laser, as indicated by the cyan dashed box. Times are min:s from the start of the recording.

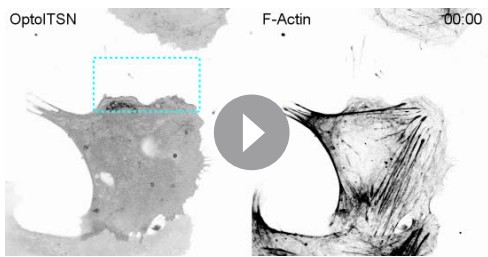

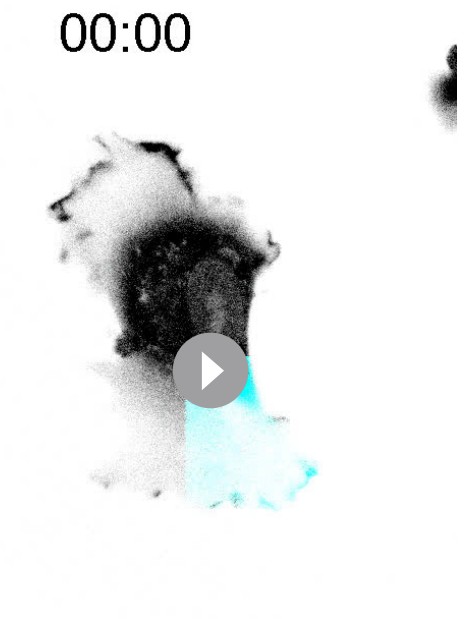

**Animation 27.** Confocal time-lapse of a blood outgrowth endothelial cell (BOEC) stably expressing Lck-mTurquoise2-iLID (not shown) and SspB-HaloTag-ITSN1(DHPH) (left) stained with JF552 nm dye. Additionally, stained for F-actin with SiR-actin (right). The cell was locally photo-activated with a 442 nm laser, as indicated by the cyan dashed box. Times are min:s from the start of the recording.

morphology, showcasing the spatial control of the optogenetic tool.

## Optogenetic tool setup

During the setup of the optogenetically recruitable GEFs we made a number of noteworthy observations. When comparing the two optogenetic heterodimerization tools iLID and eMags we found that iLID had a higher recruitment efficiency. Another study finds recruitment efficiency values in the same range but with a slightly higher recruitment efficiency for magnets, the progenitor of eMags, in comparison to iLID (*Benedetti et al., 2018*). In our study the recruitment efficiency of eMags was hampered when the iSH domain was fused to its C-terminus, consequently eMags did not induce the signaling pathway efficiently. However, another study did create a functional C-terminal fusion proteins with eMags, and magnets have been successfully applied in recruiting the iSH domain (*Benedetti et al.,*

**Animation 29.** Confocal microscopy time-lapse of a blood outgrowth endothelial cell (BOEC) stably expressing Lck-mTurquoise2-iLID (not shown) and SspB-HaloTag-TIAM1(DHPH) (gray inverted), stained with JF552 nm dye. The cell was repeatedly, locally photo-activated with a 442 nm laser line, as indicated by cyan area. Times are min:s from the start of the recording.

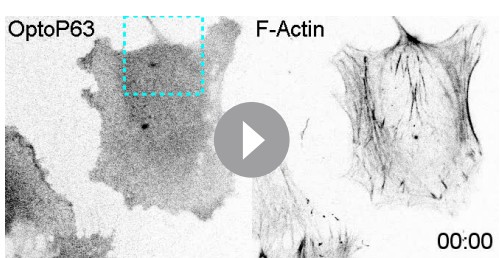

**Animation 28.** Confocal time-lapse of a blood outgrowth endothelial cell (BOEC) stably expressing Lck-mTurquoise2-iLID (not shown) and SspB-HaloTag-p63RhoGEF(DH) (left) stained with JF552 nm dye. Additionally, stained for F-actin with SiR-actin (right). The cell was locally photo-activated with a 442 nm laser line, as indicated by the cyan dashed box. Times are min:s from the start of the recording.

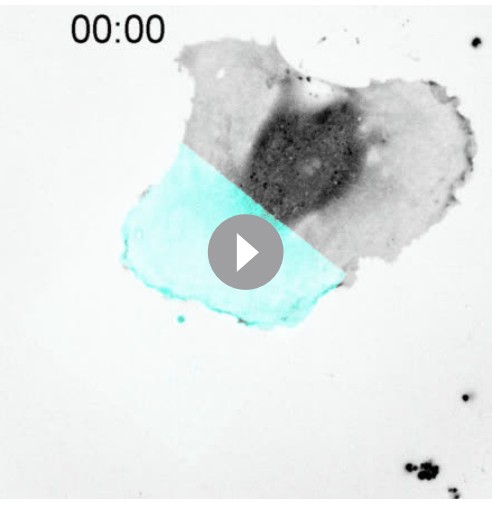

**Animation 30.** Confocal microscopy time-lapse of a BOEC stably expressing Lck-mTurquoise2-iLID (not shown) and SspB-HaloTag-TIAM1(DHPH) (gray inverted). Stained with JF552 nm dye. The cell was repeatedly, locally photo activated with a 442 nm laser line, as indicated by cyan area. Times are min:s from the start of the recording.

*2020*; *Kawano et al., 2015*). Hence, the approach appears feasible but requires some optimization, potentially the linkers. Furthermore, a correlation between recruitment efficiency and the biological response was observed, therefore the tool should be optimized for recruitment efficiency. The chemically induced rapamycin heterodimerization tool had by far the highest recruitment efficiency, however it lacks the fast reversibility and the subcellular inducibility, but it can be applied when global signaling induction is required. Another option is the BcLOV system that intrinsically gets recruited to the plasma membrane when activated by light and a tool box for GEF recruitment has been optimized recently (*Berlew et al., 2022*). However, this elegant one-component system is temperature-dependent and therefore does not allow sustained activation at 37°C and does not allow the choice of other targets, other than the plasma membrane for follow-up studies (*Benman et al., 2022*). We continued to work with the iLID tool and could improve its recruitment efficiency by using the Lck membrane tag instead of the CaaX box as a previous study suggested (*Natwick and Collins, 2021*). Furthermore, we introduced the HaloTag to the system and labeled the mobile prey component with it. The HaloTag can be stained in living cells prior to imaging, which allows flexibility in the choice of a fluorophore and the far dyes are magnitudes brighter than far red fluorescent proteins, improving the image quality (*Grimm et al., 2015*). We also found single-color location-based biosensors for Rho GTPase activity valuable to verify the Rho GTPase activation, as the common CFP-YFP FRET pair, which most biosensors utilize, is not compatible with the iLID tool, which is activated in the same excitation spectrum (*Mahlandt et al., 2021*).

Previous studies have used optogenetically recruitable GEFs for Rho GTPase activity as a proof of principle experiment in single cells to show local cell morphology changes (*Berlew et al., 2022*; *Guntas et al., 2015*; *Levskaya et al., 2009*). In a cell collective optogenetic recruitment of GEFs to induce Rho activity has been applied in the epithelium of *Drosophila* (*Herrera-Perez et al., 2021*; *Izquierdo et al., 2018*) but the potential of optogentic GEF recruitment has not been explored in the endothelium yet. Here, we demonstrate the impact of optogentic GEF recruitment in an entire monolayer and its influence on vascular barrier strength. The observed increase and decrease in vascular barrier strength induced by Opto-RhoGEFs is in line with previous research. OptoTIAM and OptoITSN activation increased barrier strength, as expected for Rac and Cdc42 activation previously shown by S1P induction, and also for photo-activated Rac1 (*Klems et al., 2020*; *Reinhard et al., 2017*). On the cell morphology level OptoTIAM induced protrusions and increased cell area, as expected for Rac activity (*Hall, 1998*). However, Cdc42 activation through OptoITSN did not induce the formation of filopodia as could be expected from previous research (*Hall, 1998*; *Ridley, 2015*) but we observed cell spreading and ruffling and the same was observed for rapamycin-induced ITSN1(DHPH) recruitment (*Reinhard, 2019*). Potentially, other signals than solely Cdc42 activity are required in endothelial cells to induce filopodia, which could be mediated by other domains of the GEF or a complex, as we apply only on the DHPH domain. Rho activation through OptoP63 decreases the vascular barrier strength as has been observed before for thrombin-induced Rho activation (*Birukova et al., 2013*; *Reinhard et al., 2016*). OptoP63-induced Rho activation caused stress fiber formation and cell contraction as reported previously (*Hall, 1998*). Previous studies lack the reversibility of the Opto-RhoGEF activation. We could observe the effect of the immediate stop of the induction of Rho GTPase activity, and noticed an immediate change in resistance after photo-activation independent of the activation length. This fast reversibility is unique to optogenetic tools and gives the opportunity to study the regulation of Rho GTPase activity in time.

Additionally, the fast reversibility might enable the study of the balance between activity of different Rho GTPases. Here, we observed the resistance dropping below baseline after photo-activation for OptoTIAM-expressing monolayers. Potentially, it is the balance between Rac and Rho signaling that is disturbed during the abrupt stop of Rac activation causing cell contraction because the balance is tilting toward Rho activity (*Burridge and Wennerberg, 2004*). However, the cells adapt quickly, likely restoring the Rac/Rho activity balance and thereby the vascular barrier strength returns to baseline levels. Interestingly, during photo-activation the resistance did plateau at a level above baseline and was not get downregulated by the reestablishment of the Rac/Rho balance. Also, the OptoP63-induced resistance decrease was counteracted, the endothelial monolayer restored the vascular barrier strength during photo-activation, suggesting the cells could restore the Rac/Rho balance. Similar observations were made for the induced changes in cell area. For example, OptoP63 induced a dramatic cell size decrease but cells start to recover already during the activation. This

shows the ability of a cell to adjust to a stimulus, and make a clear case for rapidly inducible systems like Opto-RhoGEFs to study effects in real time. On one hand, we were able to globally activate the entire monolayers on a centimeter scale in the ECIS and permeability assay with LEDs, while minimally disturbing the cells. This minimized the chance of introducing an artifact during the activation. On the other hand, we induced lamellipodia on a subcellular level with the FRAP unit at the confocal microscope on a micrometer scale. No other technique but optogenetics combines the fast reversibility, the local to global activatibility, and non-invasive activation with light. In combination with Rho GTPase activity biosensors the reversibility of the Opto-RhoGEFs provides the opportunity to study the balance between the activity of different Rho GTPases. The simultaneous application of different GEFs, individually activatable, would enable researchers to mimic the orchestra of Rho GTPase signaling better.

## Vascular barrier strength

To understand vascular barrier strength most studies focus on the junctions. However, multiple factors contribute to the vascular barrier strength, namely adherens junctions, tight junctions, focal adhesions, and potentially also cell-cell overlap. Vascular barrier studies often use knockdowns or overexpression, which the cells can adapt to, and stimuli that need to be added to the medium, causing disturbance in the ECIS resistance curve (*Birukova et al., 2013*; *Birukova et al., 2016*; *Tian et al., 2016*).

Applying Opto-RhoGEFs in endothelial monolayers we found clues that the Rho GTPase-mediated cell-cell overlap plays a bigger role in vascular barrier strength than previously anticipated. Vascular barrier strength research is presenting VE-cadherin as the central player (*Corada et al., 1999*; *Radeva and Waschke, 2018*). We showed that Opto-RhoGEF-induced Rho GTPase activity alters vascular barrier strength while VE-cadherin homotypic interactions are blocked with a blocking antibody. This effect is reversible within minutes and we observed cell-cell overlap and cell area change on that time scale in our time-lapse movies. Ideally, we would have quantified cell-cell overlap in the monolayer over time but the contrast in the images did not allow for reliable segmentation. This could be solved by mosaic expression of two different fluorophores in the monolayer. We could also not find a reliable way to measure junction properties, except for the linearity factor. Changing fluorescent intensity during the time-lapse movie and too low resolution made it challenging to measure junction properties in a quantitative manner. Junction Mapper is a good initiative to standardize junction analysis but it is not feasible for high throughput and relies on intensity (*Brezovjakova et al., 2019*). To further investigate the role of VE-cadherin, biosensors could be applied for example to monitor the phosphorylation state, as the phenotype imaged by a simple VE-cadherin stain might not reveal all the changes that influence the barrier strength.

The application of the VE-cadherin blocking antibody showed clearly that the VE-cadherin plays a role in vascular barrier strength, however the activation of Opto-RhoGEFs still altered the barrier strength in these conditions. We found that cell-cell overlap changes on the same time scale as the vascular barrier strength suggesting it is the cell-cell overlap that mediated the vascular barrier strength increase. The VE-cadherin state seems to influence the baseline of vascular barrier strength but not the amplitude by which it can be increased by the Opto-RhoGEFs. Another study has found that S1P treatment increased the vascular barrier strength while VE-cadherin heterodimerization was hampered with either EGTA, blocking antibody or siRNA and it was suggested that lamellipodia increase the vascular barrier strength (*Xu et al., 2007*). Moreover, not only S1P promotes endothelial barrier independent from VE-cadherin, also Tie-2 can increase barrier resistance in the absence of VE-cadherin (*Frye et al., 2015*). Additionally, lamellipodia have been recognized to increase vascular barrier strength and the localization of TIAM1 to the plasma membrane is associated with actin dynamics that seal intercellular gaps (*Birukova et al., 2013*; *Breslin et al., 2015*). However, S1P is also known to activate Rac and Cc42 signaling, stabilizing VE-cadherin complexes and thereby promoting barrier function (*Lee et al., 1999*; *Reinhard et al., 2017*). It has been found that VE-cadherin increasingly localizes at junctions after 10 min of S1P treatment (*Xu et al., 2007*). This increase would not explain the rapid increase in barrier strength within minutes that we observed. Nevertheless, S1P also increases cell size potentially leading to cell-cell overlap in a monolayer (*Endo et al., 2002*; *Reinhard et al., 2017*). It has been hypothesized that lamellipodia-induced cell-cell overlap proceeds the adherens junction assembly (*Jaffe and Hall, 2005*). Moreover, junction-associated intermittent lamellipodia have been observed at locations of jagged unstable VE-cadherin junctions, facilitating the

maturation of adherens junctions, thereby maintaining the vascular barrier (*Abu Taha et al., 2014*). This suggests that the relatively stable VE-cadherin junctions are complemented by dynamic lamellipodia to establish and maintain barrier integrity. Our data supports the idea that cell-cell overlap is required and already increases the barrier strength and in combination with intact VE-cadherin junctions it results in an even higher barrier strength. It would be interesting to closely examine the contribution of cell-cell overlap to the vascular barrier strength. Rac activity has been directly related to lamellipodia formation, hence increasing the cell-cell overlap. Cdc42 activity, however, has been associated with filopodia formation, which does not contribute much to the cell-cell overlap (*Hall, 1998*). Nevertheless, we observed a vascular barrier strength increase upon Cdc42 activation, if this increase was cell-cell overlap-mediated it could be explained by recently described lamellipodia-like structures induced by Cdc42 in the absence of the WAVE regulation complex (*Kage et al., 2022*).

In conclusion, junctions do play a role in vascular barrier strength. This was shown by the blocking antibody decreasing the barrier strength. However, another cell property is still able to increase this barrier strength. This may be the cell-cell overlap, as the cell size change correlates well with the barrier strength increase and the overlap appears within the time frame of barrier strength increase. Nevertheless, there are other factors that have not been studied here, for example focal adhesions influence ECIS measurements as well, or tight junctions, or other adherens junction proteins.

## Limitations and perspectives

The following paragraphs will discuss limitations of the Opto-RhoGEFs. Regarding the timing, the recruitment of Opto-RhoGEFs and the subsequent activation of Rho GTPases seems fast enough to manipulate Rho GTPase activity on a physiological time scale. We measured $t_{1/2}$ ON kinetics for iLID of 5.14 s in that time range the imaging speed at a confocal microscope rather becomes the limiting factors than the recruitment kinetics. Nocodazol induced Rho activity pulses last for minutes (*Graessl et al., 2017*) as well as S1P induced Rho GTPase activity (*Reinhard et al., 2017*). Regarding the spatial aspect, by design eMags photo-activation is better locally confined than iLID (*Benedetti et al., 2018*). However, our experiments showed local morphology changes, which were indeed not restricted to the photo-activation area but still near that area. It was possible to give cells a migration direction, hence the local activation is at least sufficient to polarize the cell into leading and rear edge. A disadvantage of the iLID system is that it occupies the blue part of the light spectrum, not allowing the use of CFP and GFP without the activation of the tool. That also excludes the use of biosensors using the CFP-YFP FRET pair but recently a optogentic compatible sYFP2-mScarlet-I FRET pair has been described (*Van Geel et al., 2020a*) and single-color location-based sensors can be applied (*Mahlandt et al., 2021*).

The Rho GTPase specificity of the Opto-RhoGEFs is defined by the utilized GEF. Biosensors could be applied to confirm the specificity. ITSN1 has been shown to be specific for Cdc42 and not activating Rac1 (*Reinhard, 2019*). TIAM1 and p63RhoGEF have been classified specific for Rac and Rho respectively (*Müller et al., 2020*). Opto-RhoGEFs for other Rho GTPases can be engineered by the same design strategies as in this study, as long as a specific GEF is available. However, it cannot be excluded that Rho GTPases activate each other by crosstalk as Cdc42 is able to activate Rac1 via Pak (*Obermeier et al., 1998*), which might mean that the OptoITSN-induced cell spreading is Rac mediated which got activated by Cdc42 crosstalk. An alternative is the design of a photo-activatable Rho GTPase, which has been achieved for Rac1 (*Wu et al., 2009*). This approach is Rho GTPase specific, however, we observed basal activity of the inactive PA-Rac1, expressing cells were larger, suggesting Rac activity. The design of these proteins is more complex than the design of Opto-RhoGEFs. Another concern is the basal activity, because the GEFs are essentially overexpressed in the cytosol. This is not the location where they activate Rho GTPases but they can freely diffuse to the plasma membrane. We attempted to assess basal activity by measurements of the cell area, because active Rho GTPases are known to alter cell size (*Klems et al., 2020*). We could not measure a change in cell size, however the different cell lines run at different resistance levels in the ECIS experiment, OptoITSN and Opto-TIAM higher than the control and OptoP63 lower, matching the expected effect of the targeted Rho GTPases. Also, in the permeability assay OptoTIAM cells kept in the dark show a somewhat lower permeability than control cells, suggesting basal activity. One reason for basal activity could be the photo-activation by daylight. However, cells were kept in the dark throughout the experiment and the iLID recruitment is reversible within 20 s. To further prevent basal activity a tool could be designed

where the components are inactive and only get activated when stimulated with blue light. One approach could be to split the DH and the PH domain, assuming they are both inactive when separated, and fuse the PH domain to iLID and the DH domain to SspB. This way the domains only become a catalytically active GEF domain once the SspB is recruited toward the iLID. Other researchers have designed a two-component system where they combined the rapamycin tool and the optogenetic heterodimerization tool TULIP to recruit dynein to Rab11 vesicles (*van Bergeijk et al., 2015*). Another approach would be to cage the catalytically active GEF domain or immobilize it at location of the cell, where it is not signaling, to be first released and recruited upon activation.

Further research could engineer Opto-RhoGEFs with other GEF specific for less well-studied Rho GTPases, as the design is rather straightforward. This will contribute to the molecular toolbox available to manipulate Rho GTPases. Furthermore, it would be of interest to separately recruit different Opto-RhoGEFs in one cell to investigate the interplay between Rho GTPases. Additionally, biosensor could be used to study the crosstalk between Rho GTPases and, if these are activated on a subcellular level, one could study how polarity during cell migration is established. The Opto-RhoGEFs could also be used to steer migration of a stem cell or a cancer cell, as we showed the potential to let a cell migrate in a directed manner. This has been done for immune cells and epithelial cell in *Drosophila* embryo (*Izquierdo et al., 2018*; *O'Neill et al., 2016*). In the long run one could imagine light-based therapy where the patient would be treated by light to control vascular barrier strength in a defined location for example to allow for more leukocyte transmigration to eliminate a tumor, or strengthen vascular barrier integrity in chronic inflammation.

The Opto-RhoGEFs have great potential to manipulate Rho GTPase signaling on a subcellular to global level in a reversible manner to study the role of Rho GTPases in the endothelium in novel ways.

## Methods
### Plasmids and cloning
Rapamycin system
C1-Lck-FRB-mTurquoise described before (*van Unen et al., 2015*) and backbone C1-sYFP1-2xNES-FKBP12, FKBP12 (*Goedhart et al., 2013*) and NES (*Kremers et al., 2006*) were described previously, were digested with BsrGI and NdeI, and ligated. The resulting C1-Lck-FRB-mTurquoise-2xNES-FKBP12 and insert IRES-sYFP2 (addgene#110623) were digested with BsrGI and ligated, creating C1-Lck-FRB T2098L-mTurquoise-IRES-sYFP2xNES-FKBP12. C1-Lck-FRB-mTurquoise2 was created by

**Table 1.** PCR primers for insert amplification.

| PCR product/insert | Template | Primer sequence |
|---|---|---|
| iSH | CFP-FKBP-iSH (addgene #20159) | FW 5'ATATTCCGGAtccaagtaccaacaagaccagg-3' |
| | | RV 5'ATAT GAATTCcgtgcgctcctcgtg-3' |
| iRFP670 | N1-iRFP670 (addgene #45457) | FW 5'- ATATGAATTCATGGCGCGTAAGGTCGAT-3' |
| | | RV 5'- ATATGGGCCCGCGTTGGTGGTGGGC-3' |
| iLID | Venus-iLID-CaaX (addgene #604110) | FW 5'-ATGGACGAGCTGTACAAGGGT-3' |
| | | RV 5'-ATACCTCGAGTTACTTTGTCTTTGACTTCTTTTTCTTCTT-3' |
| HaloTag | HaloTag (Ariana Tkachuk) | FW 5'-ATATACCGGTCGCCACCatggccgagatcggca-3' |
| | | RV 5'-ATATTGTACACgccgctgatctccagg-3' |
| Lck-mTurquoise2-iLID SspB-HaloTag-ITSN(DHPH) SspB-HaloTag-RhoGEFp63(DH) | C1-Lck-mTurqoiuse2-iLID C1-SspB-HaloTag-ITSN1(DHPH) C1-SspB-HaloTag-RhoGEFp63(DH) | FW 5'-CTTGGCAGTACATCAAGTGTATCATATGCC-3' |
| | | RV 5'-CCTCTACAAATGTGGTATGGCTGATTATGATC-3' |
| SspB-HaloTag-TIAM1(DHPH) | C1-SspB-HaloTag-TIAM1(DHPH) | FW 5'-ATATGGTACCATGAGCTCCCCGAAACG-3' |
| | | RV 5'-ATATGCGGCCGCTCACTGTCTTCTGTGTTTATCTCGC-3' |

digesting the backbone C1-Lck-FRB-mTurquoise and the insert mTurquoise2 (*Goedhart et al., 2012*) with AgeI and BsrGI and subsequently ligating the two fragments C1-mNeonGreen-FKBP12-iSH was created by ligating the AgeI and BsrGI digested backbone CFP-FKBP-iSH (addgene #20159) and the likewise digested mNeonGreen.

### eMags

eMagAF-EGFP-PM (addgene #162247) and eMagBF-TgRFPT (addgene #162253) were a gift from Pietro De Camilli. N1-iRFP670 (addgene #45457) was a gift from Vladislav Verkhusha. CFP-FKBP-iSH (addgene #20159) was a gift from Tobias Meyer. The insert iSH was PCR-amplified with primers shown in *Table 1* and digested with BspEI and EcoRI. iRFP670 was PCR-amplified with primers shown in *Table 1* and digested with EcoRI and ApaI. The backbone eMagB-TagRFP was digested with BspEI and ApaI, all three parts were ligated, creating eMagB-iSH-iRFP670.

### iLID

pLL7.0-Venus-iLID-CaaX (from KRas4B) was a gift from Brian Kuhlman (addgene #604110). ITSN-mCherry-SspB (addgene #85220) was a gift from Narasimhan Gautam. eMagB-iSH-iRFP670 was digested with KpnI and BspEI to ligate the insert iSH-iRFP670 to the likewise digested backbone ITSN-mCherry-SspB, to create iSH-iRFP670-SspB. C1-Lck-mTurquoise2-iLID was created by ligating the backbone Lck-mTurquoise2 (addgene #98822) and the PCR-amplified insert iLID, from Venus-iLID-CaaX (addgene #604110) with primers shown in *Table 1*, both digested with BsrGI and XhoI.

### Biosensors

mCherry-Akt-PH was described before (*Kontos et al., 1998*). The cysteine-free or secretory HaloTag version 7 was provided by Ariana Tkachuk in consultation with Erik Snapp (Janelia Research Campus of the Howard Hughes Medical Institute). The HaloTag was PCR-amplified with primers shown in *Table 1* and AgeI, BsrGI digested and ligated into a likewise digested C1 (Clontech) backbone, creating C1-HaloTag. C1-HaloTag-3xrGBD has been created by digesting the backbone mNeonGreen-3xrGBD (addgene# 176091) with AgeI and BsrGI and ligating it to the likewise digested insert from C1-HaloTag. CMVdel-dimericTomato-wGBD has been described elsewhere (addgene #176099).

### Opto-RhoGEFs

C1-SspB-mCherry-RhoGEFp63DH was created by digesting the backbone C1-SspB-mCherry (*Van Geel et al., 2020b*) with AgeI and BclI and ligating it to the likewise digested insert RhoGEFp63(DH) (addgene #67898). C1-SspB-mCherry-ITSN1(DHPH) was created by digesting the backbone C1-SspB-mCherry with AgeI and SacII, ligating it to the likewise digested insert mCherry-ITSN1 DHPH, as described previously (*Reinhard, 2019*). C1-SspB-mCherry-TIAM1(DHPH) was created by digesting the backbone C1-SspB-mCherry with AgeI and SacII, ligating it to the likewise digested insert mCherry-TIAM SS DHPH, as described previously (*Reinhard, 2019*).

Color variants were created by digesting the backbones C1-SspB-mCherry-RhoGEFp63(DH), C1-SspB-mCherry-ITSN1(DHPH), and C1-SspB-mCherry-TIAM1(DHPH) with AgeI and BsrGI and ligating them to the likewise digested inserts HaloTag or iRFP670.

pLV-Lck-mTurquoise2-iLID was created by digesting the pLV vector with NheI and SnaBI and the PCR-amplified insert, with primers shown in *Table 1*, from C1-Lck-mTurqoiuse2-iLID with SnaBI and XbaI and ligating the two fragments. pLV-SspB-HaloTag-ITSN1(DHPH) was created by digesting the pLV vector with NheI and SnaBI and the PCR-amplified insert, with primers shown in *Table 1*, from C1-SspB-HaloTag-ITSN1(DHPH) with SnaBI and XbaI and ligating the two fragments. pLV-SspB-HaloTag-RhoGEFP63(PH) was created by digesting the pLV vector with NheI and SnaBI and the PCR-amplified insert with primers shown in *Table 1* from C1- SspB-HaloTag-RhoGEFp63(DH) with SnaBI and XbaI and ligating the two fragments.

pLV-SspB-HaloTag-TIAM1(DHPH) was created with Lenti Gateway cloning (Invitrogen). Therefore, C1-SspB-HaloTag-TIAM1(DHPH) was PCR-amplified with the primers shown in *Table 1*, digested with KpnI and NotI and ligated to the likewise digested entry vector pENTR1A. Subsequently, this vector was recombined with the pLenti6.3/V5 DEST backbone.

The plasmid pLV-mScarlet-I-CaaX was described previously (*Arts et al., 2021*).

**Table 2.** Addgene numbers for plasmids created in this study.

| Name | Addgene number |
| --- | --- |
| iSH-iRFP670-SspB | 176102 |
| C1-Lck-mTq2-iLID | 176125 |
| C1-HaloTag-3xrGBD | 176108 |
| C1-SspB-mCherry-p63DH | 176112 |
| C1-SspB-mCherry-DHPH-ITSN | 176103 |
| C1-SspB-mCherry-DHPH-TIAM | 176104 |
| C1-SspB-HaloTag-DHPH-ITSN1 | 176113 |
| C1-SspB-HaloTag-DHPH-TIAM1 | 176114 |
| C1-SSpB-HaloTag-p63DH | 176116 |
| C1-SspB-iRFP670-DHPH-ITSN1 | 176117 |
| C1-SspB-iRFP670-DHPH-TIAM1 | 176118 |
| C1-SspB-iRFP670-p63DH | 176120 |
| pLV-Lck-mTq2-iLID | 176130 |
| pLV-SspB-HaloTag-TIAM-DHPH | 176127 |
| pLV-SspB-HaloTag-ITSN-DHPH | 176128 |
| pLV-SspB-HaloTag-p63DH | 176129 |
| pEntr1a-SspB-HaloTag-TIAM-DHPH | 176134 |

Plasmids created in this study are available on addgene, see *Table 2*.

## Cell culture

Human embryonic kidney cells 293T (HEK293T) cells (CRL-3216, American Tissue Culture Collection; Manassas, VA, USA) were cultured in Dulbecco's modified Eagle's medium+GlutaMAX (DMEM) (Gibco) with 10% fetal calf serum (Gibco) and 100 U/ml penicillin-streptomycin (Thermo Fisher Scientific) from passage number 10 to 30, at 37°C in 7% $CO_2$. HEK293T cells were used for lentiviral particle production.

Henrietta Lacks (HeLa) cells (CCL-2, American Tissue Culture Collection; Manassas, VA, USA) were cultured in Dulbecco's modified Eagle's medium+GlutaMAX (Gibco) with 10% fetal calf serum (Gibco) (DMEM+FCS) at 37°C in 7% $CO_2$, from passage number 20 to 60. cbBOECs were isolated from healthy donor umbilical cord blood as described before (*Martin Ramirez et al., 2012*). Cells were cultured in Endothelial Cell Growth Medium-2 BulletKit (CC-3162, Lonza) with 100 U/ml penicillin (Thermo Fisher Scientific) and 100 µg/ml streptomycin (Thermo Fisher Scientific) and 20% fetal calf serum (EGM-18) at 37°C in 5% $CO_2$. cbBOECs were cultured from passage number 4 to 16. Culture dishes and microscopy dishes were coated with 0.1% gelatin (CAS 9000-70-8, Merck) in phosphate-buffered saline 30 min prior to cell seeding. The morphology, growth characteristics, and response to S1P (known to increase resistance) were consistent with those of endothelial cells.

The HEK293T and HeLa cells were from a commercial provider that confirmed the identity of the cell line. All cell lines were routinely tested for *Mycoplasma* contamination.

## Stable cell lines

Lentiviral particles were produced in HEK293T cells, seeded on day 0 in a T75 culture flasks and transfected on day 1 with TransIT (Mirus) using third-generation packing plasmids (pHDMG·G VSV ENV, pHDM·HgpM2 GAG/POL, pRC-CMV-Rev1b REV) in combination with pLV-Lck-mTurquoise2-iLID, pLV-SspB-HaloTag-RhoGEFP63(DH), pLV-SspB-HaloTag-ITSN1(DHPH), pLV-SspB-HaloTag-TIAM1(DHPH), or pLV-mScarlet-CaaX. The supernatant was harvested on day 2 and 3 after transfection, filtered (0.45 µm) and concentrated using Lenti-X Concentrator (TakaraBio cat #631232), resuspended in

EGM-2, and stored at –80°C. cbBOECs seeded subconfluently in a T75 culture flask, were transduced with Lck-mTurquoise2-iLID virus, solely as a control, and in combination with either SspB-HaloTag-RhoGEFP63(DH), SspB-HaloTag-ITSN1(DHPH), or SspB-HaloTag-TIAM1(DHPH) virus. Two days after transduction, double positive cells expressing Lck-mTurquoise2-iLID and SspB-HaloTag-RhoGEFP63(PH), SspB-HaloTag-ITSN1(DHPH), or SspB-HaloTag-TIAM1(DHPH) were sorted, using a BD FACSAria cell sorter. Prior to sorting, the HaloTag was stained with JF552 nm dye. mScarlet-CaaX-transduced BOECs were selected by treatment with puromycin for 24 hr.

## Sample preparation for microscopy

For transfection 25,000–50,000 000 HeLa cells were seeded on round 24 mm ø coverslip (Menzel, Thermo Fisher Scientific) in a six-well plate with 2 ml DMEM+FCS. The transfection mix contained 1 µl linear polyethylenimine (PEI, pH 7.3, Polysciences) with a concentration of 1 mg/ml per 100 ng DNA and 0.5–1 µg plasmid DNA per well, mixed with 200 µl OptiMEM (Thermo Fisher Scientific) per well. After 15 min incubation at room temperature the transfection mix was added to the cells 24 hr after seeding. HeLa cells were imaged between 24 and 48 hr after transfection in an Attofluor cell chamber (Thermo Fisher Scientific) in 1 ml of Microscopy Medium (20 mM HEPES [pH = 7.4], 137 mM NaCl, 5.4 mM KCl, 1.8 mM $CaCl_2$, 0.8 mM $MgCl_2$, and 20 mM glucose) at 37°C.

For imaging, cbBOECs stably expressing Opto-RhoGEFs were seeded subconfluently in 3 cm ø glass bottom microscopy dishes (MakTek Corporation) or grown into a monolayer in NuncLab-Tek chambered #1.0 Borosilicate Coverglass System 8 chambers microscopy dish (Thermo Fisher). cbBOECS were imaged in EGM-18 medium at 37°C and 5% $CO_2$.

The HaloTag was stained, for at least 1 hr prior to imaging, with a concentration of 150 nM of Janelia Fluor Dyes (JF) JF552 nm (red) or JF635 nm (far red) (Janelia Materials). The medium was replaced before imaging. Rapamycin (LC Laboratories) was used at a final concentration of 100 nM for chemogenetic stimulation, directly during the imaging. VE-cadherin was stained 1:200 with the live labeling antibody Alexa Fluor 647 Mouse Anti-Human CD144 (BD Pharmingen), 2 min prior to imaging. The antibody remained in the medium, cells were not washed. F-actin was stained with SiR-actin, absorption 652 nm and emission 674 nm (SC001, Spirochrome) at a final concentration of 500 nM for at least 2 hr before imaging. The medium was replaced before imaging. The nucleus was stained with SiR-DNA, absorption 652 nm and emission 674 nm (SC007, Spirochrome) at a final concentration of 500 nM for at least 1 hr before imaging. The medium was replaced before imaging. To inhibit homotypic extracellular VE-cadherin binding, the VE-cadherin blocking antibody-purified mouse anti-cadherin-5 (BD Transduction Laboratories) was used at a final concentration of 2.5 µg/ml and incubated for at least 1 hr. PECAM was stained 1:200 with the live labeling antibody Alexa Fluor 647 Mouse Anti-Human CD31 (BD Pharmingen), 2 min prior to imaging. The antibody remained in the medium, cells were not washed.

## Spinning disk microscopy

HeLa cells were imaged with a Nikon Ti-E microscope equipped with a Yokogawa CSU X-1 spinning disk unit, a 60× objective (Plan Apo VC, oil, DIC, NA 1.4), Perfect Focus System and the Nikon NIS elements software. Images were acquired with a Andor iXon 897 EMCCD camera. Photo-activation was achieved with a single pulse of the 440 nm laser light, intensity set to 20%, for 1 s. During photo-activation CFPs were imaged using a 440 nm laser line, a triple dichroic mirror (440, 514, 561 nm), and a 460–500 nm emission filter. RFPs were imaged using a 561 nm laser line, a triple dichroic mirror (405, 488, 561 nm), and a 600–660 nm emission filter.

## Wide-field microscopy

Mosaic monolayers of mScarlet-CaaX or Lck-mTurquiose2-iLID, SspB-HaloTag-TIAM1(DHPH) expressing BOECs were imaged at a Nikon Eclipse TI equipped with a lumencor SOLA SE II light source, standard Nikon filter cubes for CFP, mCherry and Cy5, a 60×1.49 NA Apo TIRF (oil) objective, and an Andor Zyla 4.2 plus sCMOS camera (2×2 binning).

## Confocal microscopy Sp8

Confocal microscopy images were obtained at a Leica Sp8 (Leica Microsystems) equipped with a 63× objective (HC PL Apo, C2S, NA 1.40, oil) and a 40× objective (HC PL Apo, CS2, NA 1.30, oil), using

unidirectional line scan at a scan speed of 400 Hz. For HyD detectors the gain was set to 100 V unless indicated differently. Images were acquired with 1024×1024 pixel resolution and 16-bit color depth unless indicated differently. To keep the focal plane during time-lapse movies the adaptive focus system was used in continuous mode.

Heterodimerization kinetics were imaged in HeLa cells at a frame rate of 2.5 s, 512×512 pixel resolution and the pinhole was set to 1 AU. Photo-activation was achieved by scanning with a 488 nm laser line, the argon laser power set to 20%, intensity set to 1%, every 2.5 s for a total 2 min with no line averaging. During photo-activation GFP or YFP signal was detected with a HyD detector, 498–552 nm emission detection range. Sequentially, RFP signal was imaged by using a 561 nm laser line at 0.5% intensity and a HyD detector with a 570–680 nm emission detection range. For the rapamycin system CFP signal was imaged by using a 442 nm laser line, set to an intensity of 2% in combination with a HyD detector, 452–512 nm emission detection range. Sequentially, YFP signal was imaged by using a 514 nm laser line, intensity set to 1%, in combination with a HyD detector, 524–624 nm emission detection range.

PIP3 biosensor relocalization was imaged in HeLa cells at a frame rate of 10 s, the pinhole was set to 1 AU. Photo-activation was achieved by scanning with a 488 nm laser line, the argon laser power set to 20%, intensity set to 1%, every 10 s for a total 2 min, with 2× line averaging. During photo-activation GFP or YFP signal was detected with a PMT detector, 498–558 nm emission detection range, gain set to 700 V. Simultaneously, far red signal from JF635 nm dye was imaged with a 633 nm laser line, intensity set to 20%, in combination with a HyD detector with a 643–734 nm emission detection range. Sequentially, RFP signal was imaged by using a 561 nm laser line at 2% intensity and a HyD detector with a 571–630 nm emission detection range. For the rapamycin system, CFP signal was imaged by using a 442 nm laser line, set to an intensity of 2% in combination with a HyD detector, 448–486 nm emission detection range. Sequentially, GFP signal was detected by using a 488 nm laser line, intensity set to 1%, in combination with a PMT detector, 498–556 nm emission detection range, gain set to 800 V. Sequentially, RFP signal was detected, using a 561 nm laser line, intensity set to 2%, in combination with a HyD detector, 568–699 nm emission detection range.

Rho GTPase biosensor relocation was imaged in HeLa cells at a frame rate of 20 s, with the pinhole set to 1 AU and 2× line averaging. Photo-activation was achieved by scanning with a 442 nm laser, set to 1% intensity, every 20 s for a total of 280 s, and 2× line averaging. Red fluorescent signal was detected with a 561 nm laser line, intensity set to either 15% or 20%, in combination with a HyD detector, 571–630 nm emission detection range. Sequentially, far red fluorescence was imaged with a 633 nm laser line, intensity set to 5%, in combination with a HyD detector, 647–710 nm emission detection range. Sequentially, CFP was imaged during the photo-activation, using a PMT detector, with a 452–510 nm emission detection range, gain set to 700 V.

To image cbBOECs stably expressing Opto-RhoGEFs, seeded subconfluently in 3 cm ø glass bottom microscopy dishes (MakTek Corporation), the pinhole was set to 3 AU unless stated differently. CFP signal was detected during photo-activation in combination with a PMT detector, 450–509 nm emission detection range, gain set to 700 V. Simultaneously, red fluorescent signal was imaged with a 561 nm laser line, intensity set to 2%, in combination with a HyD detector, 570–633 nm emission detection range. Sequentially, far red fluorescent signal was detected with a 633 nm laser line, set to 5% intensity, in combination with a HyD detector, emission detection range 643–734 nm. Global photo-activation was achieved by scanning with the 442 nm laser line, set to 2% intensity, 2× line averaging, every 15 s for a total of 10 min. Time-lapse had a frame interval of 15 s. For local photo-activation the FRAP setup was used to photo-activate a region of 20×20 µm² with the 442 nm laser line, intensity set to 1%, every 20 s for total of 10 min, with 2× line averaging. Time-lapse had a frame interval of 20 s. Each cell was locally photo-activated twice, for the second activation the activation region was moved to the opposite side of the cell. cbBOECs additionally stained with SiR-actin were imaged under the same conditions as described above, with a frame interval of 10 s or 20 s and a total photo-activation time of either 10 min or 20 min, pinhole set to 1 AU and 2× line averaging.

cbBOECs seeded in parallel with the ECIS experiments, stably expressing Opto-RhoGEFs, in NuncLab-Tek chambered #1.0 Borosilicate Coverglass System 8 chambers microscopy dish (Thermo Fisher), at a density of 50,000 cells per chamber were grown into a monolayer. Subsequently, the monolayers were imaged with the pinhole set to 3 AU, at a frame rate of 10 s and no line averaging. Photo-activation was achieved by scanning with a 442 nm laser line, intensity set to 1%,

every 10 s for a total of 10 min (15 min OptoTIAM) repeated once with a 10 min (15 min Opto-TIAM) recovery time in between, with no line averaging. CFP fluorescence was detected during photo-activation with a HyD detector, 452–523 nm emission detection range. Simultaneously, red fluorescence was detected with a 561 nm laser line, set to 2% intensity, in combination with a HyD detector, 571–656 m emission detection range, gain set to 75 V. Sequentially, far red fluorescence was detected with a 633 nm laser line, intensity set to 10%, in combination with a HyD detector, 643–724 nm detection range. cbBOECs additionally treated with the VE-cadherin blocking antibody were photo-activated once for 15 min, at the same conditions described above. Red fluorescence was detected in the first sequence and CFP and far red signal were detected simultaneously in the second sequence. cbBOECs additionally stained for PECAM were imaged at a frame interval of 20 s with 2× line averaging otherwise at same conditions as described for the cells treated with the VE-cadherin blocking antibody.

## LED photo-activation

For global photo-activation in incubators an LED strip safety RGB LED strip Combo 12V/ 24V SMD 3528/50505 LED strip (Fuegobird) was used, set to blue light (peak 470 nm) at the highest intensity setting 9, illuminating continuously. The LED strip was taped to the lid of the cell culture dish at an approximate distance of 1 cm from the cells. The blue LED light spectrum was measured for these conditions with a spectrometer USB2000 (Ocean Optics). The intensity was measured for the wavelength of 470 nm with a power-meter PM100D (THORLABS) with an S170C sensor, sensor area $18 \times 18$ mm$^2$, at approximately 1 cm distance from the LEDs. For comparison the light intensity of the 442 nm laser of the Sp8 was measured at 1% intensity, and the 488 nm laser was measured at 20% laser power and 1% intensity. Light power density was defined by the light intensity over the illuminated area. For the LED the entire sensor of 18 mm$^2$ was illuminated. For the laser light the illuminated area was defined as the first ring of the airydisk for the $63 \times 1.4$ NA objective, which is $1.165 \ 10^{-13}$ m$^2$ for the 442 nm laser and $1.42 \ 10^{-13}$ m$^2$ for the 488 nm argon laser.

## Resistance measurement in an endothelial monolayer

To perform resistance measurements, the ECIS ZTheta (Applied BioPhysics) machine was used. Eight-well, 10-electrode (8W10E PET) ECIS arrays were pretreated with 10 mM L-cysteine (Sigma) for 15 min at 37°C, washed twice with 0.9% NaCl solution and coated with 10 µg/ml fibronectin in 0.9% NaCl solution (Sigma) for at least 1 hr at 37°C. cbBOECs stably expressing Opto-RhoGEFs were seeded at 50,000 cells per well density to grow into a monolayer. The resistance measurement at 4000 Hz, every 10 s, at 37°C and 5% CO$_2$ was immediately started after seeding. The photo-activation with LEDs was started approximately 16 hr after seeding. To inhibit homotypic extracellular VE-cadherin binding, the VE-cadherin blocking antibody-purified mouse anti-cadherin-5 (BD Transduction Laboratories) was used at a final concentration of 2.5 µg/ml in EMG-18 (control only EGM-18) and incubated for at least 1 hr before photo-activation. Roughly, 3 hr prior to the antibody addition the EGM-18 medium was refreshed for all conditions. After photo-activation the EMG-18 medium was refreshed again for all conditions. To test the resistance increase ability of the monolayer, cells were stimulated with S1P (Sigma-Aldrich) at a final concentration of 650 nM in EMG-18 (control only EGM-18).

## Permeability measurement in an endothelial monolayer

To measure permeability, cbBEOCs expressing Opto-RhoGEFs (20,000 cells per insert) were seeded on gelatin-coated 24-well cell culture inserts (Corning FluoroBlok, Falcon, 3.0 µm pore size) in a 24-well plate (Corning Companion Plate, Falcon) and grown into a monolayer. Forty-eight hr after seeding the medium was removed from the upper compartment and replaced with either FITC 0.3 kDA (Sigma), used at a final concentration of 4 µg/ml, FITC 10 kDa dextran (Sigma) or FTIC dextran 70 kDa dextran (Sigma) used at a final concentration of 833 µg/ml. Subsequently, cells were photo-activated with blue LED light for 10 min at 37°C and 5% CO$_2$. Immediately after the 10 min photo-activation, the inserts were removed from the 24-well plate and leakage was measured using an Infinite F200 pro plate reader (TECAN) at 37°C. FITC fluorescence was measured with an excitation wavelength of 500 nm (bandwidth 9 nm) and emission was detected at a wavelength of 525 nm (bandwidth 20 nm) for four positions in one well. The average of these four measurements was used for analysis.

## Analysis

FIJI (*Schindelin et al., 2012*) was used to analyze raw microscopy images. Representative microscopy images were adjusted for brightness. When the look up tables 'mpl-inferno' and 'mpl-magma' were applied, brighter colors represent higher intensity values.

To measure the membrane to cytosol intensity ratio, represented in *Figure 1A*, a membrane ROI was created based on the bait channel, therefore an overtime relatively immobile part of the membrane was selected in the first frame of activation. The ROI of the membrane was defined by thresholding based on the default FIJI settings and adjusted by hand. The ROI for the cytosol was drawn by hand. Images were background corrected and the mean gray values were measured for the defined ROIs in each frame. The ratio was calculated by membrane mean intensity value over cytosolic mean intensity value. The ratio was normalized by dividing each value by the ratio of the first frame.

The $t_{1/2}$ ON and $t_{1/2}$ OFF were estimated by fitting a monoexponential curve to the mean values of the normalized membrane to cytosol intensity ratio. The Rmarkdown file that is used for the curve fitting is available: https://doi.org/10.5281/zenodo.7152276.

To measure the change in membrane over cytosolic intensity ratio for the PIP3 location sensor, shown in *Figure 1B*, images were background-corrected. ROIs were drawn by hand for cytosol and plasma membrane based on the bait channel. The mean gray value for pre-activation was measured at time point 0 s and for activated at 50 s of activation in the defined ROIs. The ratio for membrane over cytosol mean intensity for the PIP3 sensor was calculated for the pre- and activated condition and then the pre- over activated ratio was calculated representing the change in PIP3 sensor location.

To measure the change in cytosolic intensity represented in *Figure 1D*, images were background-corrected. Cytosol ROIs were drawn by hand. The ratio of mean gray value before photo-activation to mean gray value first frame of photo-activation, for the prey channel, was calculated and presented as a percentage.

To measure the cytosolic intensity change for Rho GTPase location sensors shown in *Figure 1F and G*, images were background-corrected. dimericTomato-wGBD images were additionally bleach-corrected with the exponential fit method in FIJI. ROIs were drawn by hand for the cytosol and the mean gray intensity for these was measured over time. For plotting, the intensity values were normalized by dividing each value by the value of the first frame.

Cell area change, shown in *Figures 4A and 5A*, was depicted by overlaying the frame before photo-activation and maximum response of the cell. These time points are indicated in the legend. Images were converted to binary (0=background, 255=cell area), threshold set by FIJI default. The first frame was subtracted by the value 254, the second frame was subtracted by the value 253, and the sum of these two images was calculated. The resulting image had four intensity values: 0 representing the background, 1 representing areas of cell contraction, 2 representing areas of cell spreading, and 3 representing unchanged area. Colors for these areas were adjusted in Illustrator (Adobe). The meaning of the colors can be found in the legends.

To measure cell area and the form factor, shown in *Figure 4B* and *Figure 4—figure supplement 1B and C*, CellProfiler (Version 4.1.3) (*Stirling et al., 2021*) was used. The pipeline is included in the deposited data. Raw images were uploaded to CellProfiler. The primary object (nucleus) was identified in the SiR-DNA images for objects between 30 and 100 pixel diameter. The secondary object (entire cell) was identified in the Opto-RhoGEF channel JF552 nm red fluorescence, Gaussian filter with sigma 1 was applied before watershedding from the primary object with the Otsu two-class thresholding method. The threshold correction factor was adjusted per dataset. Objects touching the edge were excluded from the analysis. The 'EditObjectsManually' was used to correct for improperly identified cell edges. Area and form factor were measured and exported to an Excel spread sheet for these edited, secondary objects (entire cell). This method was used to measure the initial cell area in *Figure 4—figure supplement 1A*. To track the cell area and form factor change per cell shown in *Figure 4—figure supplement 1B and C* ,the frames 1, 48, and 88, that is 0 s, 705 s, and 1305 s, were analyzed, and 'TrackObjects' was added to the pipeline for the secondary object based on overlap. To track the cell area in every frame (*Figure 4B*), cells were selected, that did not touch the edge or neighboring cells, to use fully automated segmentation, so 'EditObjectManually' could be removed from the pipeline.

To measure the activation area covered by cells, photo-activated locally, FIJI was used. The first frame of activation was compared to the last frame of a 10 min photo-activation. Images were

background-corrected. The image was cropped for the photo-activation area and the Gaussian blur filter, with a sigma of 1, was applied to the Opto-RhoGEF JF552 nm channel. To identify the area covered by cell, the Huang dark threshold was applied, and if necessary adjusted by hand, to create a ROI. The area of this ROI was measured and plotted as a percentage of photo-activation area covered by the cell, pre- and post-activation. ROIs are included in the deposited data.

To measure the junction linearity index the FIJI plugin 'Tissue Analyzer' (*Aigouy et al., 2010*) was used. To segment the junctions, the watershed algorithm was used with the default parameters and strong blur set between 18 and 22, weak blur set to 1.5. When necessary the segmented junctions were corrected by hand. Based on the junction segmentation, ROIs were created. The polygon tool was used to draw a polygon per cell corner, matching points with three cell junctions, representing the minimum distance between the corners. The perimeter was measured for the junction ROI and the minimum distance polygon. The linearity index was calculated by junction perimeter divided by polygon perimeter.

To represent junctional movement, the FIJI function 'Temporal-Color-Code' was used on the same amount of frames for pre-, post-, during photo-activation images of the VE-cadherin staining. The look up table 'fire' was applied to the resulting image and inverted. Light colors represent earlier junction positions and darker colors represent later junction positions. Color code is displayed in the figure.

To measure cell-cell overlap area, binary images were created by applying the filter Gaussian blue (sigma 2) and a threshold (Huang). The resulting 8-bit binary images for the CFP channel were subtracted by 254 and the RFP image by 254. Subsequently, the two images were added up and the areas with the value 3 were used as ROIs to measure the overlap area. The data was normalized by division through the initial value for the overlap area.

## Plots and statistics

Dot plots were generated with the R-based web tool 'PlotsofData' and 'SuperPlotsofData' (*Goedhart and Pollard, 2021*; *Postma and Goedhart, 2019*). Plots over time were generated with the R-based web tool 'PlotTwist' (*Goedhart, 2020*). Figures were made in Illustrator (Adobe). Statistical analysis was performed in the R-based 'PlotsofData' and 'PlotTwist' web tools (*Goedhart, 2020*; *Postma and Goedhart, 2019*). The 95% confidence intervals were calculated by bootstrapping.

## Acknowledgements

We want to thank Ronald Breedijk for the support at the van Leeuwenhoek Centre for Advanced Microscopy, Section Molecular Cytology, Swammerdam Institute for Life Sciences, University of Amsterdam. We also want to thank Anne-Marieke van Stalborch, Jos van Rijssel, Max Grönloh, and Werner van der Meer for their support and advice on the work with endothelial cells, the permeability assay, and ECIS experiment. Funding: EM was supported by an Nederlandse Organisatie voor Wetenschappelijk Onderzoek ALW-OPEN grant (ALWOP.306).

## Additional information

### Funding

| Funder | Grant reference number | Author |
| --- | --- | --- |
| Nederlandse Organisatie voor Wetenschappelijk Onderzoek | ALWOP.306 | Jaap D van Buul Joachim Goedhart |
| ZonMw | VICI grant #91819632 | Jaap D van Buul |

The funders had no role in study design, data collection and interpretation, or the decision to submit the work for publication.

### Author contributions

Eike K Mahlandt, Conceptualization, Data curation, Formal analysis, Supervision, Investigation, Visualization, Methodology, Writing - original draft, Project administration, Writing - review and editing;

Sebastián Palacios Martínez, Formal analysis, Investigation, Visualization; Janine JG Arts, Resources; Simon Tol, Investigation, Methodology; Jaap D van Buul, Conceptualization, Supervision, Funding acquisition; Joachim Goedhart, Conceptualization, Software, Supervision, Funding acquisition, Visualization, Project administration, Writing - review and editing

**Author ORCIDs**
Eike K Mahlandt http://orcid.org/0000-0001-9458-8543
Sebastián Palacios Martínez http://orcid.org/0000-0001-6626-5798
Jaap D van Buul http://orcid.org/0000-0003-0054-7949
Joachim Goedhart http://orcid.org/0000-0002-0630-3825

Reviewer #1 (Public Review): https://doi.org/10.7554/eLife.84364.3.sa1
Reviewer #2 (Public Review): https://doi.org/10.7554/eLife.84364.3.sa2
Reviewer #3 (Public Review): https://doi.org/10.7554/eLife.84364.3.sa3
Author Response: https://doi.org/10.7554/eLife.84364.3.sa4

## Additional files

### Supplementary files
• MDAR checklist

### Data availability
The data generated during this study is available at Zenodo.org: https://doi.org/10.5281/zenodo.7944191.

The following dataset was generated:

| Author(s) | Year | Dataset title | Dataset URL | Database and Identifier |
|---|---|---|---|---|
| Mahlandt E, Martínez P, Arts S, Janine JG, Tol S, van Buul JD, Goedhart J | 2022 | Opto-RhoGEFs: an optimized optogenetic toolbox to reversibly control Rho GTPase activity on a global to subcellular scale, enabling precise control over vascular endothelial barrier strength | https://doi.org/10.5281/zenodo.7944191 | Zenodo, 10.5281/zenodo.7944191 |

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
