## [Editor Report · eLife assessment]

This paper presents a **valuable** advance in the ability to manipulate the integrity of the barrier between endothelial cells. A wide range of data are presented, offering **convincing** support for the effectiveness of the method. This work is likely to attract a diverse audience of both cell biologists and researchers developing tools to manipulate cell and tissue function.

---

## [Referee Report · Reviewer #1 (Public Review)]

This manuscript by Mahlandt, et al. presents a significant advance in the manipulation of endothelial barriers with spatiotemporal precision, and in the use of optogenetics to manipulate cell signaling in vascular biology more generally. The authors establish the role of Rho-family GTPases in controlling the cytoskeletal-plasma membrane interface as it relates to endothelial barrier integrity and function, and adequately motivate the need for optogenetic tools for global and local signaling manipulation to study endothelial barriers.

Throughout the work, the optogenetic assays are conceptualized, described, and executed with exceptional attention to detail, particularly as it relates to potential confounding factors in data analysis and interpretation. Comparison across experimental setups in optogenetics is notoriously fraught, and the authors' control experiments and measurements to ensure equal light delivery and pathway activation levels across applications is very thorough. In demonstrating how these new opto-GEFs can be used to alter vascular barrier strength, the authors cleverly use fluorescent-labeled dextran polymers of different sizes and ECIS experiments to demonstrate the physiological relevance of BOEC monolayers to in vivo blood vessels. Of particular note, the resiliency of the system to multiple stimulation cycles and longer time course experiments is promising for use in vascular leakage studies.

Given that dozens of Rho GTPase-activating GEFs exist, expanded rationale for the selection of p63, ITSN1, and TIAM1 in the form of discussion and literature citations would be helpful to motivate their selection as protein effectors in the engineered tools. Extensive tool engineering studies demonstrate the superiority of iLID over optogenetic eMags or rapamycin-based chemogenetic tools for these purposes. However, as the utility of iLID and eMags has been demonstrated for manipulation of a variety of signaling pathways, the iSH-Akt demonstration does not seem necessary for these systems.

The demonstration of orthogonality in GTPase- and VE-cadherin-blocking antibody-mediated barrier function decreases and is compelling, even without full elucidation of the role of cell size or overlap in barrier strength. The discussion section presents a mature and thoughtful description of the limitations, remaining questions, and potential opportunities for the tools and technology developed in this work. Importantly, this manuscript demonstrates a commitment to scientific transparency in the ways in which the data are visualized, the methods descriptions, and the reagent and code sharing it presents, allowing others to utilize these tools to their full potential.

---

## [Referee Report · Reviewer #2 (Public Review)]

This manuscript reports on the use of Optogenetics to influence endothelial barrier integrity by light. Light-induced membrane recruitment of GTPase GEFs is known to stimulate GTPases and modulate cell shape, and here this principle is used to modulate endothelial barrier function. It shows that Rac and CDc42 activating constructs enhance barrier function and do this even when a major junctional adhesion molecule, VE-cadherin, is blocked. Activation of Rac and Cdc42 enhanced lamellipodia formation and cellular overlaps, which could be the basis for the increase in barrier integrity.

The authors aimed at developing a light driven technique with which endothelial barrier integrity can be modulated on the basis of activating certain GTPases. They succeeded in using optogenetic tools that recruit GEF exchange domains to membranes upon light induction in endothelial cell monolayers. Similar tools were in principle known before to modulate cell shape/morphology upon light induction, but were used here for the first time as regulators of endothelial barrier integrity. In this way it was shown that the activation of Cdc42 and Rac can increase barrier integrity even if VE-cadherin, a major adhesion molecule of endothelial junctions, is blocked. Although it was shown before that stimulation of S1P1 receptor or of Tie-2 can enhance endothelial barrier integrity in dependence of Cdc42 or Rac1 and can do this independent of VE-cadherin, the current study shows this with tools directly targeting these GTPases.

Furthermore, this study presents very valuable tools. The immediate and repeatable responses of barrier integrity changes upon light-on and light-off switches are fascinating and impressive. It will be interesting to use these tools in the future in the context of analyzing other mechanisms which also affect endothelial barrier function and modulate the formation of endothelial adherens junctions.

---

## [Referee Report · Reviewer #3 (Public Review)]

Mahlandt et al. report the design and proof of concept of Opto-RhoGEF, a new set of molecular tools to control the activation by light of the three best known members of the Rho GTPase family, RhoA, Rac1 and Cdc42.

The study is based on the optogenetically-controlled activation of chimeric proteins that target to the plasma membrane guanine nucleotide exchange factors (GEFs) domains, which are natural activators specific for each of these three Rho GTPases. Membrane-targeted GEFs encounter and activate endogenous Rho proteins. Further investigation on the effect of these tools on RhoGTPase signaling would have strengthened the report.

These three Opto-RhoGEFs are reversible and enable the precise spatio-temporal control of Rho-regulated processes, such as endothelial barrier function, cell contraction and plasma membrane extension. Hence, these molecular tools will be of broad interest for cell biologists interested in this family of GTPases.

Mahlandt et al. design and characterize three new optogenetic tools to artificially control the activation of the RhoA, Rac1 and Cdc42 by light. These three Rho GTPases are master regulators of the actin cytoskeleton, thereby regulating cell-cell contact stability or actin-mediated contraction and membrane protrusions.

The main strength of this new experimental resource lies in the fact that, to date, few tools controlling Rho activation by reversibly targeting Rho GEFs to the plasma membrane are available. In addition, a comparative analysis of the three Opto-RhoGEFs adds value and further strengthens the results, given the fact that each Opto-GEF produces different (and somehow expected) effects, which suggest specific GTPase activation. The design of the tools is correct, although the membrane targeting could be improved, since the Lck N-terminus used to construct the recombinant proteins contains myristoylation and palmitoylation sites, which has the potential to target the chimeric protein to lipid rafts. As a consequence, this may not evenly translocate these Rho-activating domains.

An additional technical feature that must be highlighted is an elegant method to activate Opto-RhoGEFs in cultured cells, independent of laser and microscopes, by using led strips, which notably expands the possibilities of this resource, potentially allowing biochemical analyses in large numbers of cells.

The experimental evidence clearly indicates that authors have achieved their aim and designed very useful tools. However, they should have taken more advantage of this remarkable technical advance and investigate in further detail the spatiotemporal dynamics of Rho-mediated signaling. Although the manuscript is a "tool and resource", readers may have better grasped the potential benefits of tuning GTPase activity with this tool by learning about some original and quantitative insights of RhoA, Rac1 and Cdc42 function.

One of such insights may have come from the set of data regarding the contribution of adherens junctions. The effect of other endothelial cell-cell junctions, such as tight junctions, may also contribute to barrier function, as well as junctional independent, cell-substratum adhesion. These optogenetic tools will undoubtedly impact on these future studies and help decipher whether these other adhesion events that are important for endothelial barrier integrity are also under control of these three GTPases. Overall, the manuscript is sound and presents new and convincing experimental strategies to apply optogenetics to the field of Rho GTPases.

---

## [Author Response]

The following is the authors' response to the original reviews.

We were pleased with the overall enthusiastic comments of the reviewers:

Reviewer #1: “This manuscript by Mahlandt, et al. presents a significant advance in the manipulation of endothelial barriers with spatiotemporal precision”Reviewer #2: “The immediate and repeatable responses of barrier integrity changes upon light-on and light-off switches are fascinating and impressive.”Reviewer #3: “, these molecular tools will be of broad interest to cell biologists interested in this family of GTPases.”

We thank the reviewers for their fair and constructive comments that helped us to improve the manuscript.

**Reviewer #1 (Recommendations For The Authors):**
1. This paper is likely to attract a diverse audience. However, the order of data presented in this manuscript can be confusing or challenging to follow for the naive reader. This is because the tool characterization is split into two parts: before the barrier strength assay (selection of optogenetic platform and tool expression) and after (characterization of cell morphology with global and local optogenetic stimulation). Reorganizing the results such that the barrier strength results follows from an understanding of individual cell responses to stimulation may improve the ability of this readership to understand the factors at play in the changes in barrier strength observed when opto-RhoGEFs are activated.

We appreciate this idea, and we initially structured the paper in the proposed order and then decided, that we wanted to put more focus on the barrier strength results by already presenting them in the second figure. Therefore, we prefer to keep this order of figures.

1. While the description of the selection of iLID as the study's optogenetic platform is clear, a better job could be done motivating the need for engineering new optogenetic tools for the control of GEF recruitment. Given that iLID-based tools for GEFs of RhoA, Rac1, and Cdc42 already exist, some of which are cited in the introduction, more information on why these tools were not used would be helpful-were these tools tested in endothelial cells and found lacking.

The original system has the domain structure DHPH-tagRFP-SspB. But we wanted to work with a SspB-FP-GEF construct, which would allow easy exchange of the FP and the DHPH domain. This modular approach allowed us to generate and compare the mCherry, iRFP647 and HaloTag version. We don’t want to claim that we engineered an entirely new optogenetic tool but rather optimized an existing one with different tags. To make this more clear we added : ‘*The membrane tag of the original iLID was changed to an optimized anchor. In addition, we modified the sequence of the domains to SspB, tag, GEF to simplify the exchange of GEF and genetically encoded tag. A set of plasmids with different fluorescent tags was created for more flexibility in co-imaging*.’

1. Comment on the reason behind using DHPH vs. DH domains for each GEF is needed.

We have previously found (and this is supported by biochemical analysis of GEF activity) that the selected domains provide the best activity. We will add reference and the following to the text: ‘*Their catalytic active DHPH domains were used for ITSN1 and TIAM1 (Reinhard et al., 2019). In case of p63 the DH domain only was used, because the PH domain of p63 inhibits the GEF activity (Van Unen et al., 2015) (Fig. 1E).*’

1. Since multiple Rho GTPases (e.g., RhoA, RhoB, RhoC) exist and Rho is used as the name of the GTPase family, please use RhoA where applicable for clarity.

Since the RhoGEFp63 will activate RhoA/B/C we would rather not refer to RhoA only. We will clarify this in the text: ‘*Three GEFs were selected, ITSN1, TIAM1 and RhoGEFp63, which are known to specifically activate respectively Cdc42, Rac and Rho and their isoforms*.’

1. A brief comment on the use of HeLa cells for protein engineering and characterization (versus the endothelial cells motivated in the introduction) may be helpful.

We added the following to the text: ‘*HeLa cells were used for the tool optimization because of easier handling and higher transfection rate in comparison to endothelial cells.*’

Minor suggestions:In figure 1C, line sections showing intensity profiles before and after protein dimerization might further emphasize the change in biosensor localization.

We are not a fan of intensity profiles as the profile depends strongly on the position of the line and it basically turns a 2D image in 1D data, for a single image. So, we prefer to stick to the quantification as shown in panel 1B (which shows data from multiple cells).

**Reviewer #2 (Recommendations For The Authors):**
1)The study has analyzed the effects of light-induced activation of the three optogenetic constructs in endothelial cells on their barrier function (electrical resistance) at high cell density and correlated the findings with the cellular overlap-producing effects on endothelial cells cultured at sparse cell density. It should be tried to show these effects at a cell density where these light-induced effects increase electrical resistance. Lifeact with different chromophores in adjacent cells might be useful.

We had attempted to measure the overlap in a monolayer by taking advantage of the Halotag and the variety of dyes available by staining one pool of cells red with JF 552 nm and the other far red with the JF 635 nm dye. However, the cells need at least 24 h to form a monolayer and by then they had exchanged the dye and red and far red pool could not be distinguished any longer.

Therefore, we used the Lck-mTq2-iLID construct, which already marks the plasma membrane of the cells. We created a mosaic monolayer of cells expressing mScarlet-CaaX and cells expressing Lck-mTq2-iLID + SspB-HaloTag-TIAM(DHPH). We observed and increase in the overlap between cells under this condition. The results have been added to figure 4 - figure supplement 2I&J. To the text we added:

*'Additionally, cell-cell membrane overlap increased about 20 %, up on photo-activation of OptoTIAM, in a mosaic expression monolayer (figure 4 - figure supplement 2I,J, Animation 22)‘*

1. The authors correctly state that some reports have shown that S1P can increase endothelial barrier function in VE-cadherin independent ways and these are related to Rac and Cdc42. This was also shown for Tie-2 in vitro and even in vitro in the absence of VE-cadherin and should also be mentioned.

We added the following to the text: ‘*Not only S1P promotes endothelial barrier independent from VE-cadherin, also Tie2 can increase barrier resistance in the absence of VE-cadherin (Frye et al. 2015).*’

Since a blocking antibody against VE-cadherin was used, a negative control antibody should be tested which also binds to endothelial cells.

To visualize the cell-cell junctions in the experiment shown in Supplemental Fig 3.1, we added a non-blocking VE-cadherin antibody that is directly labeled with ALEXA 647 and shows normal junction morphology. These experiments already give an indication that the live labeling antibody of VE-cadherin does not disturb the junction morphology. However, when we added the blocking antibody against VE-cadherin, known to interfere with the trans-interactions of VE-cadherin, a rapid disruption of the junctions is observed.

Additionally, previous work has shown, that VE-cadherin labeling antibody does not interfere with junction dynamics and function (*see Figure 2.A, Kroon et al. 2014 ‘Real-time imaging of endothelial cell-cell junctions during neutrophil transmigration under physiological flow’, jove.*). We have added the figures below, showing that addition of the control IgG and VE-cadherin 55-7H1 Abs at the timepoint where the dotted line is, did not interfere with the resistance whereas the blocking Ab drastically reduced resistance. We have added this reference to the results. ‘*Previous work has shown the specific blocking effect of this antibody in comparison to the VE-cadherin (55-7H1) labeling antibody (Kroon et al., 2014)*.’

**Author response image 1. sa4fig1:** 

**Reviewer #3 (Recommendations For The Authors):**
Additional comments for the authors:1. The introduction is very long and would benefit from a more concise emphasis on the information required to put the work and results in context and understand their importance.

Comment: we appreciate the comment of the reviewer. However, we wish to introduce the topic and the tools thoroughly and therefore we chose to keep the introduction as it is.

1. The N-terminal membrane-binding domain does not homogeneously translocate to the plasma membrane, since lck is a raft-associated kinase. Please comment on this.

In our hands, the Lck is among the most selective and efficient tags for plasma membrane localization (https://doi.org/10.1101/160374). We do observe homogeneous translocation, but our resolution is limited to ~200 nm and so we cannot exclude that the Lck concentrates in structures smaller than 200 nm. Given the robust performance of the lck-based iLID anchor in the optogenetics experiments, we think that the Lck anchor is a good choice.

1. Figure 1D is not very clear. What does 25 or 36% change mean? If iLID tg is conjugated to these sequences, its cytosolic localization should be reduced versus iLID alone. Is this what the graph wants to express? If so, please, label properly the ordinate axis in the graph (% of non-tagged iLID values?)

The graph is representing the recruitment efficiency of SspB to the plasma membrane for the two different membrane tags, targeting iLID to the plasma membrane. The recruitment efficiency was measured by the depletion of SspB-mScarlet intensity in the cytosol, up on light activation, and represented as a change in percentage.

We added the following to the title of the graph_: SspB recruitment efficiency for Plasma Membrane tagged iLID._

1. Supplemental figures in the main text. Fig S1D in the text refers to data in Fig S1E and Fig S1E is supposed to be Fig S1F? (page 11).

That is correct. The mistakes have been corrected (and this is now renamed to figure 1 - figure supplement 1E and 1F).

1. Figure 3. Contribution of VE-cadherin. Other junctional complexes, such as tight junctions may also intervene. However, these results would also suggest that cell-substrate adhesion rather than cell-cell junctions may modulate the barrier properties, as it has been previously demonstrated for example by imatinib-mediated activation of Rac1 (Aman et al. Circulation 2012). The ECIS system used to measure TEER in the quantitative barrier function assays can modulate these measurements and discriminate between paracellular permeability (Rb) and cell-substrate adhesion (alpha). Please, provide whether the optogenetic modulation of these GTPases does indeed regulate Rb or alpha.

The measured impedance is made up of two components: capacitance and resistance. At relatively high AC frequencies (> 32,000 Hz) more current capacitively couples directly through the plasma membranes. At relatively low frequencies (≤ 4000 Hz), the current flows in the solution channels under and between adjacent endothelial cells’ (https://www.biophysics.com/whatIsECIS.php).

Therefore, the high frequency impedance is representing cell-substrate adhesion whereas the low frequency responds more strongly to changes in cell-cell junction connections.

We only measured at 4000 Hz, representing the paracellular permeability. We chose a single frequency to maximize time resolution.

We have added this extra comment to the legend of the figure: ‘*(B) Resistance of a monolayer of BOECs stably expressing Lck-mTurquoise2-iLID, solely as a control (grey), and either SspB-HaloTag-TIAM1(DHPH)(purple)/ ITSN1(DHPH) (blue) or p63RhoGEF(DH) (green) measured with ECIS at 4000 Hz, representing paracellular permeability, every 10 s.*’